

# Assessing the robustness of Antarctic temperature reconstructions over the past two millennia using pseudoproxy and data assimilation experiments

François Klein[1], Nerilie J. Abram[2,3], Mark A. J. Curran[4,5], Hugues Goosse[1], Sentia Goursaud[6,7], Valérie Masson-Delmotte[6], Andrew Moy[4,5], Raphael Neukom[8], Anaïs Orsi[6], Jesper Sjolte[9], Nathan Steiger[10], Barbara Stenni[11,12], and Martin Werner[13]

[1]Georges Lemaître Centre for Earth and Climate Research (TECLIM), Earth and Life Institute (ELI), Université catholique de Louvain (UCL), Belgium
[2]Research School of Earth Sciences, Australian National University, Canberra ACT 2601, Australia
[3]ARC Centre of Excellence for Climate Extremes, Australian National University, Canberra ACT 2601, Australia
[4]Australian Antarctic Division, 203 Channel Highway, Kingston, Tasmania 7050, Australia
[5]Antarctic Climate & Ecosystems Cooperative Research Centre, University of Tasmania, Hobart 7001, Australia
[6]Laboratoire des Sciences du Climat et de l'Environnement (IPSL/CEA-CNRS-UVSQ UMR 8212), CEA Saclay, 91191 Gif-sur-Yvette CEDEX, France
[7]Université Grenoble Alpes, Laboratoire de Glaciologie et Géophysique de l'Environnement (LGGE), 38041 Grenoble, France
[8]University of Bern, Oeschger Centre for Climate Change Research & Institute of Geography, 3012 Bern, Switzerland
[9]Department of Geology – Quaternary Science, Lund University, Sölvegatan 12, 223 62, Lund, Sweden
[10]Lamont-Doherty Earth Observatory, Columbia University, Palisades, New York, USA
[11]Department of Environmental Sciences, Informatics and Statistics, Ca' Foscari University of Venice, Venice, Italy
[12]Institute for the Dynamics of Environmental Processes, CNR, Venice, Italy
[13]Alfred Wegener Institute, Helmholtz Centre for Polar and Marine Research, 27570 Bremerhaven, Germany

**Correspondence:** François Klein (francois.klein@uclouvain.be)

**Abstract.** The Antarctic temperature changes over the past millennia remain more uncertain than in many other continental regions. This has several origins: 1) the number of high resolution ice cores is small, in particular on the Antarctic Plateau and in some coastal areas in East Antarctica; 2) the short instrumental records limit the calibration period for reconstructions and the assessment of the methodologies; 3) the link between isotope records from ice cores and local climate is usually complex and dependent on the spatial and time scales investigated. Here, we use climate model results, pseudoproxy and data assimilation experiments to assess the potential of reconstructing the Antarctic temperature over the last two millennia based on a new database of stable oxygen isotopes in ice cores compiled in the framework of Antarctica2k (Stenni et al., 2017). The well-known covariance between $\delta^{18}$O and temperature is reproduced in the two isotope-enabled models used (ECHAM5/MPI-OM and ECHAM5-wiso), but is generally weak over the different Antarctic regions, limiting the skill of the reconstructions. Furthermore, the strength of the link displays large variations over the past millennium, further affecting the potential skill of temperature reconstructions based on statistical methods which rely on the assumption that the last decades are a good estimate for longer temperature reconstructions. Using a data assimilation technique allows in theory taking into account changes in the $\delta^{18}$O-temperature link through time and space. Pseudoproxy experiments confirm the benefits of using



data assimilation methods instead of statistical ones that provide reconstructions with unrealistic variances in some Antarctic subregions. They also confirm that the relatively weak link between both variables leads to a limited potential for reconstructing temperature based on $\delta^{18}$O. The reconstruction skill is however higher and more uniform among reconstruction methods when the reconstruction target is the Antarctic as a whole rather than smaller Antarctic subregions. This consistency between the

methods at the large scale is also observed when reconstructing temperature based on the real $\delta^{18}$O regional composites of Stenni et al. (2017). In this case, temperature reconstructions based on data assimilation confirm the long term cooling over Antarctica during the last millennium, and the later onset of anthropogenic warming compared to the simulations without data assimilation, especially visible in West Antarctica. Data assimilation also allows reconciling models and direct observations by reconstructing the East-West contrast regarding the recent temperature trends, indicating that internal variability likely

plays a major role in driving this heterogeneous recent warming. This is further supported by the large spread of individual PMIP/CMIP model realizations regarding the recent warming pattern. As in the pseudoproxy framework, the reconstruction methods perform differently at the subregional scale, especially in terms of the variance of the produced time series. While the potential benefits of using a data assimilation method instead of a statistical one have been highlighted in a pseudoproxy framework, the instrumental series are too short to confirm it in a realistic setup.

## 1 Introduction

Over the last few decades, the Antarctic Peninsula and West Antarctica have experienced a strong warming while no significant temperature trend has been recorded in East Antarctica (Nicolas and Bromwich, 2014; Jones et al., 2016). The attribution of the causes of these signals is complicated by the large interannual to multi-decadal variability that characterizes the Antarctic climate (Schneider et al., 2006; Goosse et al., 2012; Jones et al., 2016), stressing the importance of considering a longer period

to put the recent changes in a wider perspective. This is not possible using the instrumental record that generally goes back to the late 1950s in Antarctica (Nicolas and Bromwich, 2014; Jones et al., 2016). Yet temperature information on a longer time scale can be inferred from stable isotope ratios of oxygen and of hydrogen recorded in ice cores (e.g. Dansgaard, 1964; Jouzel, 2003; Masson-Delmotte et al., 2006).

However, the spatial coverage of high resolution (annual to decadal) cores is uneven with a very small number in dry regions

such as the central Antarctic Plateau (Stenni et al., 2017), and in some coastal areas as in Adélie Land (Goursaud et al., 2017). In addition to the limitations related to the number and distribution of the cores, there are several sources of uncertainty in reconstructions based on those data. The period available to calibrate the records is very short due to the limited availability of instrumental records. Besides, it has been long established that the relationship between isotopes and surface temperature may differ spatially and temporally (e.g. Jouzel et al., 1997), as a result of changes in the origin of moisture, atmospheric transport

pathways (e.g. Schlosser et al., 2004), or in precipitation seasonality (e.g. Sime et al., 2008; Masson-Delmotte et al., 2008; Sodemann and Stohl, 2009). Finally, non-climatic noise related to postdepositional effects associated for instance with wind scouring and water vapour make the interpretation of ice core signals further challenging (e.g. Ekaykin et al., 2014; Ritter et al., 2016; Casado et al., 2016).

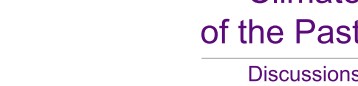

Individual ice core records may thus be affected by non-climatic, very local processes. Combining individual records in a given location or region has the potential to improve the signal to noise ratio. This has been done at the continental scale in Goosse et al. (2012) where a composite of last millennium Antarctic temperature was presented. This composite is based on the average of seven temperature records derived from isotope measurements in ice cores, and shows a weak multi-centennial

cooling trend over the pre-industrial period followed by a warming after 1850 CE. A similar last millennium cooling is observed in the reconstruction of PAGES 2k Consortium (2013) that is based on a composite-plus-scaling (CPS) approach (e.g. Schneider et al., 2006), using 11 records, but no clear recent warming is observed in this reconstruction.

Those continental-scale trends based on a limited number of records actually mask important spatial variations as shown in Stenni et al. (2017). Using a new database compiled in the framework of the PAGES Antarctica2k working group containing

112 isotopic records, Stenni et al. (2017) have produced temperature reconstructions over the last two millennia on both regional and continental scales. Those reconstructions confirm the last millennium cooling over Antarctica, which is strongest in West Antarctica, and show no evident warming in the last century at the continent-scale, despite the significant positive temperature trends observed in the Antarctic Peninsula, the West Antarctic Ice Sheet (WAIS) and coastal Dronning Maud Land (DML).

In contrast, climate model simulations performed in the framework of the third phase of the Past Model Intercomparison

Project (PMIP3; Otto-Bliesner et al., 2009) and the fifth phase of the Coupled Model Intercomparison Project (CMIP5; Taylor et al., 2012) show a general twentieth century warming over Antarctica due to the anthropogenic forcing (Goosse et al., 2012; PAGES 2k–PMIP3 group, 2015). As pointed out by Jones et al. (2016), this model-data mismatch at the continental scale suggests that either the CMIP5 models overestimate the forced response, or that the forced changes in the real world are overwhelmed by natural variability, or a combination of both. Part of the disagreement may also be due to observational gaps

and uncertainties in regional temperature estimates.

To complement the limited information available from direct temperature observations, it is necessary to extract as much reliable temperature information as possible from ice core water stable isotope records. In this context, our goal here is to assess the robustness of Antarctic temperature reconstructions over the last two millennia presented in Stenni et al. (2017), focusing on the potential impact of spatial and temporal changes in the $\delta^{18}$O-surface temperature relationship on the reconstruction

skill. This is achieved by means of model results analysis, pseudoproxy experiments and data assimilation.

Our study being based on model results, it is important to first characterize the similarities and differences between simulated, reconstructed and observed temperature over the last millennium, at the regional scale. The simulated temperature products are derived from model simulations following the Past Model Intercomparison Project (PMIP3) and the Coupled Model Intercomparison Project (CMIP5) protocols (Schmidt et al., 2011; Taylor et al., 2012), and from simulations from two

isotope-enabled climate model, ECHAM5/MPI-OM (Werner et al., 2016) and ECHAM5-wiso Werner et al. (2011). They are compared to the regional temperature reconstructions from Stenni et al. (2017), and to the instrumental-based reconstruction produced by Nicolas and Bromwich (2014), covering the recent period from 1958 CE, and defined at a 60 km resolution.

The potential of reconstructing surface temperature based on water stable isotopes is then assessed in two stages. First, through the study of the stability of the relationship between those two variables in the model world over the last millennium

using the results of the ECHAM5/MPI-OM and ECHAM5-wiso, and second, using pseudoproxy experiments. Pseudoproxy




experiments consist in using climate model results to evaluate the performance of paleoclimate reconstruction methods in a flexible and controlled framework (e.g. Smerdon, 2012). The methodologies applied to obtain the paleoclimate reconstructions are applied in the model world, where all variables are known, allowing assessing precisely and quantitatively the skill of the reconstructions. The resulting findings may not be fully valid for real-world implications, due to model biases, unrealistic

pseudoproxies or dependency of the pseudoproxy to the model from which they originate (Smerdon et al., 2016). However, the lack of real-world data, especially in Antarctica, limits or even prevents the extent of the evaluation of reconstruction methods, showing the interest of using pseudoproxy experiments. The complexity of the $\delta^{18}$O-temperature, potentially limiting the reconstruction skill, further stresses the interest of using such experiments. Here, the pseudoproxies are derived from an ECHAM5/MPI-OM simulation covering the period 800-1999 CE (Sjolte et al., 2018). Those pseudoproxies are used as input

data for the different statistical methods used in Stenni et al. (2017) for reconstructing temperature based on $\delta^{18}$O, as well as for data assimilation experiments.

When applied to palaeoclimatology, data assimilation aims at combining information from model results and proxy-based reconstructions to find estimates of past climate changes (e.g. Widmann et al., 2010; Hakim et al., 2016). Reconstructing temperature based on $\delta^{18}$O data using a data assimilation method has potentially several advantages compared to using the

statistical methods. First, data assimilation does not rely on a constant and stable relationship between $\delta^{18}$O and surface temperature, unlike the statistical methods used in Stenni et al. (2017). Second, data assimilation takes into account the spatial dependency of temperature which is not the case of the reconstructions of Stenni et al. (2017) and third, if a skilful reconstruction can be achieved, other climate variables than temperature can be reconstructed without any spatial or temporal gap, which may help interpreting the signals present in ice core records, although this falls out of the scope of the present study.

After the assessment of the statistical and data assimilation methods in the light of pseudoproxies experiments, the real temperature reconstructions over the last two millennia based on the new $\delta^{18}$O database presented in Stenni et al. (2017) are compared. Comparing the output of the different reconstruction methods allows assessing how robust the reconstructions are, and how sensitive they are to the specificities of the methods. A particular focus is placed on whether the simulated spatial pattern of recent temperature trends on the one hand and the measured and observed trends on the other can be reconciled

through data assimilation, or whether there is a fundamental discrepancy between model and data in this regard.

This study is structured as follows. The climate model simulations, the water stable isotopes records and the different reconstruction methods are described in Section 2. The simulated and reconstructed last millennium temperature changes are analyzed in Section 3, and compared to instrumental records over the recent past. The potential of reconstructing surface temperature based on water stable isotopes is assessed in Section 4, focusing on the $\delta^{18}$O-surface temperature relationship and

pseudoproxy experiments. Finally, the temperature reconstructions based on various reconstructions methods are presented and discussed in Section 5, before the conclusion.



## 2  Data and methods

### 2.1  Climate model simulations

Simulations performed with two isotope-enabled general circulation models (GCM), ECHAM5/ MPI-OM (Werner et al., 2016) and ECHAM5-wiso (Werner et al., 2011), are analyzed and used as a basis for the data assimilation experiments.

ECHAM5/MPI-OM is a fully coupled ocean–atmosphere–sea ice–land surface GCM. The simulation used in the present study covers the period 800-1999 CE with a horizontal resolution of 3.75° by 3.75° (Sjolte et al., 2018). It is driven through the past millennium by both natural and anthropogenic forcings as described in Sjolte et al. (2018). ECHAM5-wiso is an atmosphere-only GCM. The run employed in this study was performed by Steiger et al. (2017) and spans the years 1871-2011 CE at 1.125° spatial resolution. It is forced through this period by monthly historical sea ice and sea surface temperatures (SST) from the Met Office Hadley Centre's sea ice and sea surface temperature data set Rayner et al. (2003). The evaluation of those simulations against recent observations (global network of isotopes in precipitation (GNIP), IAEA/WMO (2018)) shows a relatively good agreement between simulated and observed oxygen ratios in precipitation regarding various diagnostics, including spatial patterns, magnitude of the changes, and $\delta^{18}$O–surface temperature relationships (Werner et al., 2016; Steiger et al., 2017). Focusing on Antarctica, ECHAM5/MPI-OM and ECHAM5-wiso simulate similar absolute values and spatial patterns of $\delta^{18}$O (Fig. A1) than another simulation of ECHAM5-wiso simulation nudged to ERA-Interim atmospheric reanalyses (Dee et al., 2011) over the period 1979-2013 CE Goursaud et al. (2018). This latter simulation has been extensively studied in Goursaud et al. (2018), where they concluded that, despite an overall underestimation of isotopic depletion by ECHAM5-wiso compared to a collection of water stable isotope measurements, the model correctly captures the spatial gradient of annually averaged $\delta^{18}$O data, validating the use of the model to study water stable isotopes in Antarctic precipitation. It gives confidence in using the longer simulations from ECHAM5/MPI-OM and ECHAM5-wiso for our analysis. In this study, model outputs are processed to calculate precipitation-weighted $\delta^{18}$O at the temporal resolution of the corresponding analysis for comparison with ice core records.

In addition to ECHAM results, last millennium temperature fields simulated by six models following the PMIP3 (Otto-Bliesner et al., 2009) and CMIP5 (Taylor et al., 2012) protocols are analyzed in Section 3. Details on models used in this study are listed on Table 1. See Klein et al. (2016) for a description of the forcings driving those simulations.

### 2.2  Water stable isotopes records

The data used in this study to assess and constrain model results consists in composites of water stable isotopes for seven climatically distinct regions covering the Antarctic continent (Stenni et al., 2017): East Antarctic Plateau, Wilkes Land Coast, Weddell Sea Coast, Antarctic Peninsula, WAIS, Victoria Land Coast-Ross Sea and DML Coast. Those regions, described in detail in Stenni et al. (2017), display relatively homogeneous characteristics in terms of regional climate and snow deposition processes, and were validated and refined by spatial correlation of temperature using the instrumental-based reconstruction of Nicolas and Bromwich (2014). The regional composites are based on 112 individual ice core water stable isotope records compiled in the framework of the PAGES Antarctica2k working group. Most of those records are oxygen isotope ratios, and



**Table 1.** Modeling centers, parameters and references of the climate models used in this study.

| Model name | Institution | Atmos reso | Ensemble members | | Period | Reference |
|---|---|---|---|---|---|---|
| **Isotope-enabled models** | | | | | | |
| ECHAM5/MPI-OM | Max Planck Institute for Meteorology | 48*96 | 1 | | 800-1999 | Werner et al. (2016) |
| ECHAM5-wiso | Max Planck Institute for Meteorology | 160*320 | 1 | | 1871-2011 | Werner et al. (2011) |
| **PMIP/CMIP models** | | | | | | |
| | | | past1000 | historical | | |
| CCSM4 | National Center for Atmospheric Research | 192*288 | 1 | 6 | 850-2005 | Gent et al. (2011) |
| CESM1 | National Center for Atmospheric Research | 96*144 | 10 | 10 | 850-2005 | Otto-Bliesner et al. (2016) |
| GISS-E2-R | NASA Goddard Institute for Space Studies | 90*144 | 1 | 6 | 850-2005 | Schmidt et al. (2014) |
| IPSL-CM5A-LR | Institut Pierre-Simon Laplace | 96*96 | 1 | 5 | 850-2005 | Dufresne et al. (2013) |
| MPI-ESM-P | Max Planck Institute for Meteorology | 96*192 | 1 | 2 | 850-2005 | Stevens et al. (2013) |
| BCC-CSM1-1 | Beijing Climate Center, China Meteorological Administration | 64*128 | 1 | 3 | 850-2005 | Wu et al. (2014) |

the ones that are deuterium isotopes ($\delta$D) were converted to a $\delta^{18}$O equivalent by dividing by 8, representing the slope of the global mean meteoric relationship of oxygen and deuterium isotopes in precipitation (Stenni et al., 2017).

Most individual records have a data resolution ranging from 0.025 to 5 years. In order to limit the influence of non-climatic noise induced by postdepositional processes (e.g. Münch et al., 2017; Jones et al., 2017; Laepple et al., 2018), they were all 5-year averaged for reconstructing the last two centuries and 10-year averaged for reconstructing the last two millennia. This lower temporal resolution also limits the potential influence of small age uncertainties. The spatial distribution of the individual records is strongly heterogeneous (Stenni et al., 2017). To avoid an over-representation in the composites of the areas with a relatively higher density of ice core network compared to other regions, the number of records was reduced based on a 2° latitude by 10° longitude grid in which multiple records falling into the same grid cell were averaged. This cuts down the number of individual series to 40 with the following distribution: 15 for the East Antarctic Plateau, two for the Wilkes Land Coast, one for the Weddell Sea Coast, four for the Antarctic Peninsula, ten for the WAIS, three for the Victoria Land Coast-Ross Sea sector and five for the DML Coast. In Stenni et al. (2017), different methods have been used to produce the seven composites from the preprocessed data, including a simple average per subregion, and weighted averages based on the



temperature regressions of each site and the relevant region. Here, the composites using the latter method is used. However, all methods give consistent $\delta^{18}$O trends and variability.

## 2.3 Data assimilation method

The data assimilation method used to perform the temperature reconstruction is based on a particle filter (e.g. van Leeuwen,
2009) that is applied offline, meaning that data assimilation makes use of an existing and fixed ensemble of simulated climate states. Offline data assimilation methods contrasts with online methods where the ensemble is generated sequentially, depending on the analysis made through the data assimilation process on the previous time step. An online method can theoretically outperform an offline one when the data assimilated involves a long-term trend since some components of the climate system can propagate information forward in time from one assimilation step to the next one (Pendergrass et al., 2012; Matsikaris
et al., 2015). This has been highlighted in Goosse (2017) where the oceans predominate. It also has the advantage to provide reconstructions that are consistent with changes in forcing since the temporal consistency is kept. However, online methods are computationally very expensive, which limits the ensemble size especially when using complex and high resolution models. Furthermore, previous works have shown that offline methods can be adequate and provide skilful data assimilation-based reconstructions using various kind of data, for instance surface temperature-related, hydroclimatic-related, or even sea surface
temperature-related data (e.g. Steiger et al., 2014; Hakim et al., 2016; Klein and Goosse, 2018; Steiger et al., 2018). Moreover, Steiger et al. (2017) have recently performed successfully offline data assimilation experiments based on Kalman filtering (Kalnay, 2003) using $\delta^{18}$O in ice core records. Finally, an offline data assimilation method allows using here the results from the isotope-enabled models ECHAM5/MPI-OM (Werner et al., 2016) and ECHAM5-wiso (Steiger et al., 2017) (see Section 2.1) which simulate explicitly the $\delta^{18}$O in precipitation leading to a straightforward and direct comparison in the data assimila-
tion process. This is a clear advantage over inferring the model $\delta^{18}$O based on a linear-univariate fit with local temperature as done for instance in Goosse et al. (2012), given the non-linearity and non-stationarity of the link between stable oxygen ratios and surface temperature (e.g. Masson-Delmotte et al., 2008).

The particle filter used here is implemented in the same way as in Klein and Goosse (2018) where it is detailed, so only a short description follows. For each assimilation time step (yearly, see Section 2.4), every member of the ensemble of sim-
ulated climate states is compared to data. Since only one simulation exists for ECHAM5/MPI-OM and ECHAM5-wiso, the ensembles are produced in both cases by selecting all simulated years, which makes ensembles containing 1200 members for ECHAM5/MPI-OM and 141 members for ECHAM5-wiso. The model-data comparison is performed using anomalies over the whole period covered by the simulations in order to remove any potential model biases and to focus on the variability. Based on this comparison, the likelihood, a measure of the ability of the different members to reproduce the signal recorded in the
data, is computed taking into account the uncertainties of the data. A weight proportional to the likelihood is then attributed to each member, which allows computing the weighted mean for each assimilation time step and provide the reconstruction.





## 2.4 Experimental design for data assimilation experiments

The potential of reconstructing the Antarctic surface temperature based on the assimilation of the seven subregional composites of $\delta^{18}O$ is first assessed in a controlled framework using pseudoproxy experiments. In this case, pseudoproxy are generated to match as closely as possible the real data of Stenni et al. (2017), described in Section 2.2. First, ECHAM5/MPI-OM $\delta^{18}O$

results over the grid cells containing real ice core records are extracted. As some records fall within a same grid cell, the total number of independent series is reduced from 112 to 52. The precipitation-weighted annual means of $\delta^{18}O$ are then computed from those time series, over which a Gaussian white noise is added in order to end up with signal-to-noise (SNR) ratios of 0.5. This value is commonly used to produce pseudoproxies (Smerdon, 2012), although it reaches the upper range of an average annual mean proxy (Wang et al., 2014). However, since the noise is applied directly on the measured quantity ($\delta^{18}O$) and not

on the climatic interpretation inferred from proxy (temperature), it seems adequate. To match the real data temporal resolution, the time series are 10- and 5-year-averaged over the period 800-1800 CE and 1800-2000 CE, respectively. The 52 time series are then assigned to one of the seven regions and a weight, proportional of the observed surface temperature relationship of the individual record with the corresponding region, is attributed to each pseudoproxy, in the same way as in Stenni et al. (2017). A weighted average by subregion is then performed to produce the seven pseudoproxy composites. Lastly, a temporal mask is

applied on the seven time series to match the same time coverage than the reconstructions of in Stenni et al. (2017).

The most natural choice would be to apply the same 10- and 5-year averaging procedure to the model results. However, this drastically reduces the number of the ensemble size and thus the range of possible climate states, limiting the data assimilation-based reconstructions skill. Hence, the pseudoproxy are linearly interpolated at the annual resolution which allows setting the assimilation frequency to annual and thus using all available individual years in the two models to build the ensembles, as

mentioned in the previous section. The assimilation process is done separately for the two model ensembles. The ensembles consist of the seven subregional time series for each year of the simulations, produced by averaging the precipitation-weighted annually averaged results in every grid cells of each region. Note that in the case of the assimilation of pseudoproxy derived from ECHAM5/MPI-OM using the ECHAM5/MPI-OM model ensemble, the model result corresponding to the same year than the pseudoproxy assimilated is excluded from the ensemble at each data assimilation step.

Data assimilation requires an estimate of observation uncertainties. In the case of the pseudoproxy experiments, the uncertainty of the seven time series corresponds to the variance of the noise added in the generation of those data, ranging from 0.15‰ for the East Antarctic Plateau to 0.47‰ for the Weddell Sea Coast. Unfortunately, it is not possible to determine accurately the uncertainty associated with real data. Several experiments have been performed using different estimates of observational errors to assess the sensitivity of our results to this parameter. For instance, the error has been assumed to be spa-

tially coherent (using values of 0.15‰, 0.25‰ or 0.50‰), estimated as proportional of the data series variances, as inversely proportional to the number of individual data series contained in a same model grid cell, and a combination of the above. Those different estimates of the data uncertainty have an impact on the results, but no best choice could be determined based on our experiments. Consequently, a temporally and spatially constant estimate for the data uncertainty equal to 0.25‰ has been preferred over more complex choices that may be hard to justify given the subjective considerations implied.



## 2.5 Statistical reconstructions methods

The data assimilation-based temperature reconstructions are compared in Sections 4.2 and 5 to the statistical reconstructions presented in Stenni et al. (2017), including the previous Antarctica2k temperature reconstruction published by PAGES 2k Consortium (2013) based on the CPS approach. Each of these reconstruction methods is applied to the same input data, whether the pseudoproxy or the real data consisting of the ice core records collection of Stenni et al. (2017).

The reconstruction methods developed in Stenni et al. (2017) are based on linear regressions of ice core $\delta^{18}O$ with local surface temperature on the regional average products. In the first method, the regional isotope composites were scaled based on the annual $\delta^{18}O$-temperature slopes inferred from a ECHAM5-wiso simulation nudged to ERA-Interim atmospheric reanalyses (Goursaud et al., 2018), over the period 1979-2013 CE. Given the limited length of the simulation, Stenni et al. (2017) considered annual mean anomalies to compute the slopes and applied those slopes on the 5- and 10-year binned composites. In the second approach, the normalized records were scaled to the instrumental period temperature variance at the regional scale, computed over the period 1960-1990 CE for the 5-year-binned averages and the period 1960-2010 CE for the 10-year-binned averages. Note that here, in the pseudoproxy framework, the scaling is based on the pseudoproxy of temperature over the period 1950-2000 for both averages.

A similar scaling is used in the CPS method used in PAGES 2k Consortium (2013) and applied to the larger Antarctic ice core database in Stenni et al. (2017). The CPS regional reconstructions consist of weighted averages of the normalized records falling in the subregions, the weights being based on the correlation between the records and the corresponding regional temperature time series over the period 1961-1991 CE. The composites are then scaled to the mean and the variance of the observations over the same period. Compared to the two previous statistical approaches, the CPS method has the limitation to discard more than half of the records, because it is limited to the sites where there is an overlap between the $\delta^{18}O$ records and direct temperature observations.

Each of those reconstructions rely on the same assumption that the instrumental period is representative for longer-term climate variability, which is not the case in data assimilation. They are referred in the following of the manuscript to as the statistical reconstructions.

## 3   Reconstructed and simulated last millennium temperature changes

At the continental scale, PMIP/CMIP models show a long-term cooling in Antarctica between the beginning of the last millennium and 1850 CE (Fig. 1), consistently with the results of previous model-based (Goosse et al., 2012; PAGES 2k–PMIP3 group, 2015) and observation-based (Goosse et al., 2012; PAGES 2k Consortium, 2013; Stenni et al., 2017) studies. The decrease in temperature in the reconstructions of Stenni et al. (2017) between 850 CE and 1850 CE reaches -0.62°C (based on the linear trend) in average over the three proposed reconstructions, which is in the range of the PMIP/CMIP models whose simulated cooling lies between -0.01°C for GISS-E2-R to -0.72°C for BCC-CSM1-1. The model ECHAM5/MPI-OM is the only one that does not simulate this last millennium cooling but rather a weak warming over the period.



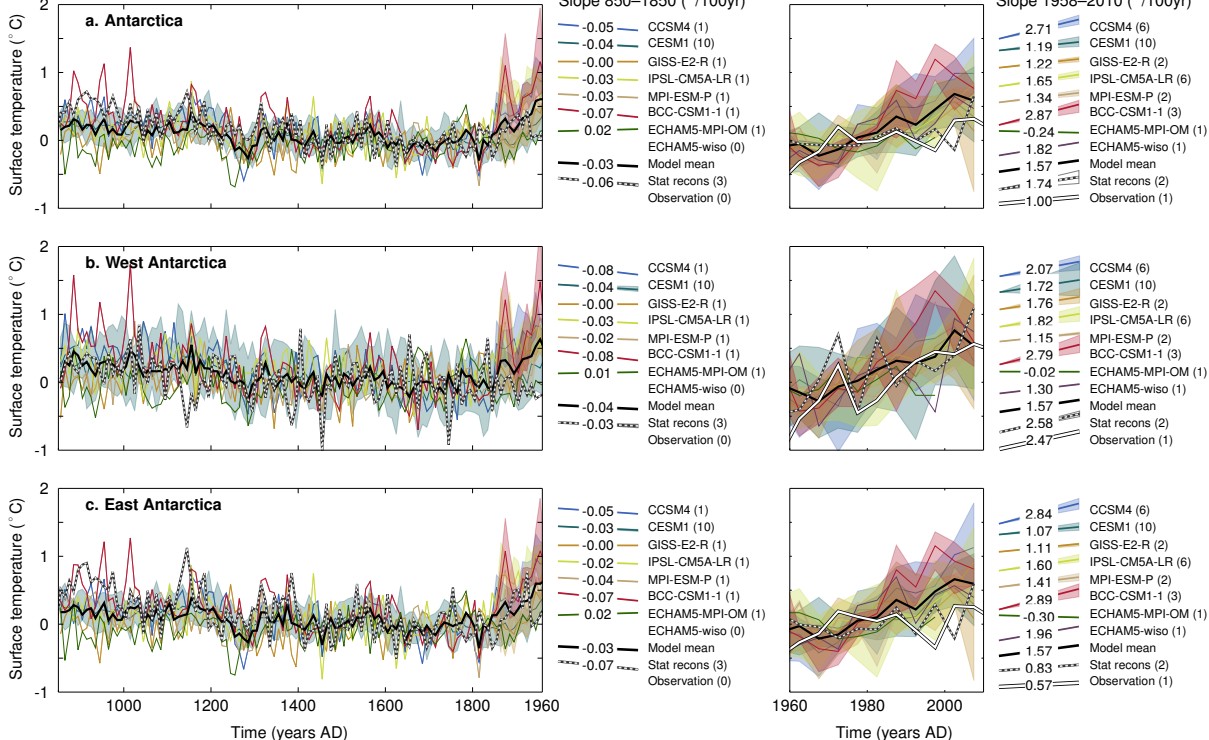

**Figure 1.** Last millennium 10- (left panels) and 5-year (right panels) mean temperature changes over a. Antarctica, b. West Antarctica and c. East Antarctica (for a definition of the regions see Stenni et al. (2017)) in models (colored lines), in the average over the three statistical reconstructions of Stenni et al. (2017) (dashed black and white line) and in the reconstruction based on instrumental observations (Nicolas and Bromwich, 2014, white line, only in right panels). The number of ensemble members of each model is given in brackets after the labels. The shaded areas represents the mean ± 1 standard deviation of the corresponding ensemble. The reference period is 1500-1800 CE for the left panels, and 1960-1990 CE for the right panels. The mean slopes (in °/100 years) over the periods 850-1850 CE and 1958-2010 CE are shown in the right of each panel, with drawn slopes proportional to the numbers.

The last millennium cooling trend is followed from the mid-nineteenth century by a relatively strong warming trend until present-day (Fig. 1). As discussed in Abram et al. (2016), climate models systematically simulate an onset of the anthropogenic warming in the mid-nineteenth century, which is consistent with palaeoclimate records in the Northern Hemisphere but not in the Southern Hemisphere where the warming is delayed. This model-data mismatch is observed in Antarctica using the

5 reconstructions of Stenni et al. (2017), especially in the western part of the continent where the delay in the reconstructions compared to models reaches about 100 year (Fig. 1-b). For the recent period where instrumental observations are available (1958-2012 CE), PMIP/CMIP models and the reconstructions slightly overestimate the observed warming, with an increase in temperature of 0.82°C for the model mean and of 0.90°C for the mean of the reconstructions based on ice core records, compared to 0.52°C for instrumental observations (Fig. 1). In contrast to other models and observations, ECHAM5/MPI-OM

10 shows a weak cooling trend over the past 50 years.



Much of the recent warming over Antarctica in the reconstruction based on instrumental observations is due to the strong increase in temperature in West Antarctica, while the East has only weakly warmed over the last 50 years. This East-West difference is also present in the statistical reconstructions, but not in the model mean that shows an uniform warming over both regions due to anthropogenic forcing (Fig. 2-a). The model mean can be seen as the forced response since the natural variability

is removed, as far as the ensemble size is large enough. As a consequence, based on a similar diagnostic, Smith and Polvani (2017) has attributed the spatial pattern to natural variability, using the model simulations from 40 CMIP5 models. The large spread between models and particularly between ensemble members of a same model confirm the role of natural variability in driving the recent temperature trend. For instance, one of the simulation of CESM1 simulates an increase in temperature of more than 2°C in West Antarctica over 1958-2005 CE, while another show a cooling trend for the same period. As a

consequence, some individual runs of CESM1, but also of IPSL-CM5A-LR and GISS-E2-R, simulate a differential warming in Antarctica resembling the observed one, with a larger temperature increase in the West Antarctica than in the East, showing that apparent model-data differences for the recent warming pattern in Antarctica do not imply a fundamental inconsistency between models and data. However, despite these important variations within the individual simulations, most of them still simulate a too homogeneous response over Antarctica compared to instrumental observations.

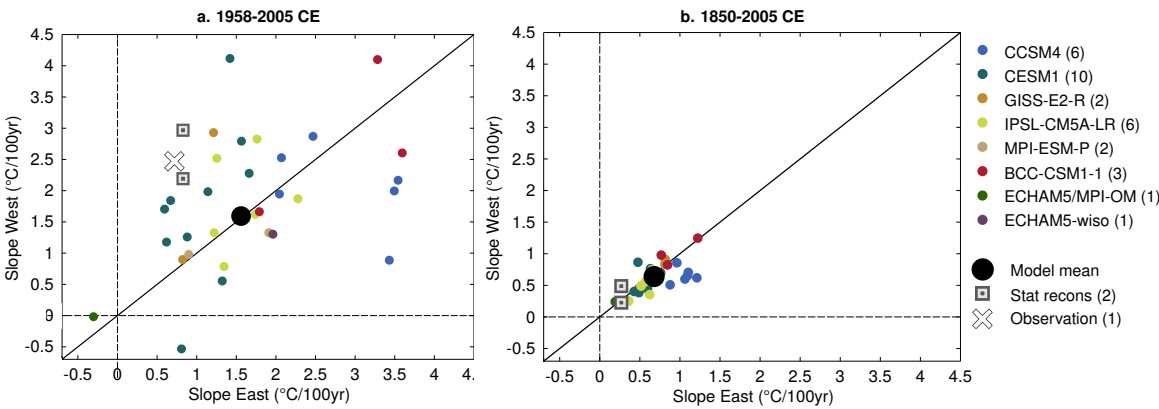

**Figure 2.** Comparison between the reconstructed, simulated and observed surface temperature slopes (in °C/100 year) in West Antarctica (y axis) and in East Antarctica (x axis), over the period a. 1958–2005 CE and b. 1850–2005 CE.

It is worth noting that values of the slopes are sensitive to the choice of the interval chosen. Shifting the period by five years forward and backyard can already lead to a difference, although the conclusions drawn from the values of the slopes hold (not shown). In contrast, the picture gets different when looking at the longer period 1850-2005 CE (Fig. 2-b). In this case, consistently with one of the two statistical reconstructions, almost all individual realizations of the models are situated on the diagonal line representing equal trends in East and West Antarctica, with a much reduced spread. Except for ECHAM5/MPI-

OM, all models, however, overestimate the warming in both regions compared to the statistical reconstructions. Unfortunately, the period covered by the instrumental record is too short to confirm or reject this uniform warming.





The next step involves investigating whether the uniform response shown by the majority of the models during the observation period 1958-2005 CE is related to a general overestimation of the correlations between Antarctic regions compared to the reconstructions based on instrumental data. In order to have a more comprehensive analysis of the link between Antarctic regions, the seven subregions defined in Stenni et al. (2017) are considered here in addition to the wider East and West Antarctica domains. In the reconstruction based on instrumental data, the link between annual mean surface temperature over East Antarctica and West Antarctica is relatively weak -although statistically significant-, with a correlation coefficient reaching 0.39 (Fig. 3). This is consistent with the difference observed in trends. This lack of strong relationship is mainly due to the Antarctic Peninsula, incorporated in West Antarctica, that appears isolated from the rest of Antarctica. This is also the case, although to a lesser extent, for the Weddell Sea Coast that is included in East Antarctica.

The simulated link between East and West Antarctica, as deduced from correlation coefficients, is rather consistent for each model mean, and comparable to the observed one. There is one exception with the model BCC-CSM1-1 that clearly overestimates the correlation, partly due to a very strong and homogeneous warming over Antarctic, BCC-CSM1-1 simulating the highest increase in temperature over the last 50 years. CCSM4 and CESM1, containing 6 and 10 ensemble members, show a relatively wide range among members that contains the observed values, with correlation coefficients between both regions varying from 0.30 and 0.65 and from 0.11 and 0.69, respectively. In contrast, all six individual members of IPSL-CM5A-LR simulate quite similar relationships between temperature over the East and the West Antarctica, ranging from 0.45 to 0.62.

The correlations between the other subregions are generally of the same magnitude in the models and the observations. Only one clear bias is observed on the link between the Weddel Sea Coast and the Antarctica as a whole, the relationship being overestimated by all individual simulations of climate models. This is related to a simulated warming in this region while the reconstruction based on instrumental observations shows no clear trend. Out of the eight models used, only BCC-CSM1-1 and ECHAM5/MPI-OM are systematically biased with a too homogeneous and heterogeneous surface temperature over Antarctica, which is partly related to their overestimated and underestimated recent warming, respectively.

Generally, there are thus no clear and systematic bias in the magnitude of the simulated correlations between subregions of Antarctica. Some models show differences with data, but there is no common rule towards an overestimated or underestimated link, while virtually all individual model representations show a more homogeneous trend in East and West Antarctica compared to data over the instrumental period 1958-2005 CE (Fig. 2-a). Obviously, there is a link between correlations between subregions and similarity of the simulated trends. For instance, the highest (lowest) correlation coefficients between East and West Antarctica simulated by BCC-CSM1-1 (ECHAM5/MPI-OM) coincide with the highest (lowest) trend values. Moreover, the wide range of trends simulated by the ensemble members of CESM1 comes together with a wide range of correlations between the two regions. However, the picture is not straightforward and highly model-dependent, rejecting the idea that the spatial coherency of the simulated trends over Antarctica can be explained only by a common interannual variability of temperature among Antarctic regions.





**Figure 3.** Simulated and observed Pearson correlation coefficients between mean annual surface temperature over 1958-2005 CE in 10 different subregions covering Antarctica (Plateau, Wilkes Land Coast, Weddell Sea Coast, Antarctic Peninsula, WAIS, Victoria Land-Ross Sea, DML coast, West Antarctica, East Antarctica and Antarctica as a whole). The order displayed for the models and the observation values is shown below the main panel. The number of ensemble members for each model is indicated in brackets. For each model, the mean of the correlations over all the members is shown, together with the value of the ensemble member that has the highest (max) and the lowest (min) correlation. The presence of a white circle represents combinations for which the null hypothesis of no correlation can be rejected at the 5% level. In the case of the mean value, there is the white circle if the majority of the ensemble members are statistically significant. The observation is the reconstruction based on instrumental observations of Nicolas and Bromwich (2014).

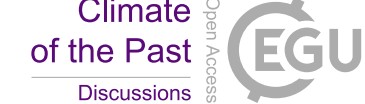

## 4   Potential of reconstructing surface temperature based on water stable isotopes

### 4.1   Relationship between $\delta^{18}$O and surface temperature in models over the last millennium

The potential of reconstructing surface temperature based on water stable isotopes is first assessed by examining the correlation coefficients and the slopes between $\delta^{18}$O and surface temperature variations over the periods covered by the isotope-enabled models used. In order to resemble the temporal resolution of the data while having a statistically significant analysis, the relationship between $\delta^{18}$O and temperature is studied using 5 year bins. At the continental scale, the correlation coefficient between $\delta^{18}$O and surface temperature simulated by ECHAM5/MPI-OM reaches 0.57 and the slope $0.78°C‰_o^{-1}$ over the period 800-1999 CE (Fig. 4-10). The $\delta^{18}$O-surface temperature relationship is not spatially homogeneous. The Antarctic Peninsula and the Victoria Land/Ross Sea sector are characterized by particularly low correlation coefficients, with values of 0.43 and 0.45, respectively. In contrast, the Plateau shows the strongest relationship with a correlation between both variables reaching 0.66. The mean slopes over the last millennium are more uniform among regions with values ranging between about $0.7-0.9°C‰_o^{-1}$, with the exception of the Victoria Land-Ross Sea region where the slope drops to 0.43 $°C‰_o^{-1}$.

The $\delta^{18}$O-temperature link varies greatly through time as can be seen when considering 100 year intervals (Fig. 4). At the continental scale, there is a strong decline of the $\delta^{18}$O-temperature link between 1600-1700 CE, where the correlation coefficient drops to 0.18 (not significant at the 0.05% level). The other centuries of the millennium are characterized by correlation coefficients varying from 0.52 to 0.69 for the periods 1000-1100 CE and 1200-1300 CE, respectively. A similar variability is observed for the slopes, with values between $0.12°C‰_o^{-1}$ in 1600-1700 CE and $1.21°C‰_o^{-1}$ in 1200-1300 CE. This variability in the diagnostics does not seem to be linked to the mean climate in a systematic way, but rather appears to be random.

The temporal variability in the correlations and the slopes is even higher at the regional scale. As for Antarctica, each sub-region has experienced over the past millennium both centuries with non significant relationships and centuries characterized by a strong $\delta^{18}$O-temperature link. The relative temporal variation of the slopes and the correlation coefficients over the last millennium strongly differs among regions, suggesting no clear imprint of the forcings on the $\delta^{18}$O-temperature link. Sime et al. (2008) have shown that the $\delta^{18}$O-temperature relationship is reduced when the climate is warmer on Antarctica, based on $CO_2$ increase simulations using the isotope enabled-version of the model HadAM3. This is not observed in the results with a last century in the range of what is simulated over the past millennium by ECHAM5/MPI-OM, but this was expected given the very limited recent warming shown by the model (Fig. 1).

Over the twentieth century, the simulated $\delta^{18}$O-temperature link between ECHAM5/MPI-OM and ECHAM5-wiso are of the same order of magnitude at the continental scale, with a correlation coefficient of 0.69 versus 0.60 and a slope of $0.57°C‰_o^{-1}$ versus $0.52°C‰_o^{-1}$ for ECHAM5/MPI-OM and ECHAM5-wiso, respectively. However, it becomes strongly different at the regional scale, with usually higher correlation coefficients and slope simulated by ECHAM5-wiso. Those values cannot be directly compared to the ones obtained using another simulation of ECHAM5-wiso nudged to ERA-Interim atmospheric reanalyses from 1979-2013 CE (Goursaud et al., 2018) used in Stenni et al. (2017), given the different time coverage and resolution (1-year mean instead of 5-year mean). However, it is striking that the slopes are systematically higher in the latter simulation.

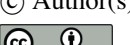



**Figure 4.** Upper panels: evolution of 5-year averaged $\delta^{18}O$ in precipitation (blue) and of surface temperature (red) over 800-1999 CE in ECHAM5/MPI-OM and over 1871-2011 CE in ECHAM5-wiso (lighter colors). Lower panels: Pearson correlation coefficients (black) and slope (in $°C‰^{-1}$) between the two variables in ECHAM5/MPI-OM (horizontal solid bars) and in ECHAM5-wiso (horizontal dashed bars). The length of the bars correspond to the period over which the diagnostics are computed. The black bars filled with white mean that the correlation is not statistically significant at the 5% level. The diagnostics were only computed over the period 1900-2000 CE for ECHAM5-wiso. The crosses represent the correlation coefficients and the slopes (in $°C‰^{-1}$) between the same variables based on another ECHAM5-wiso simulation, over the period 1979-2013 CE, using annual mean values (Stenni et al., 2017).



This is especially the case in the regions Wilkes Land Coast and Antarctic Peninsula, where the slopes reach $1.91°C‰_o^{-1}$ and $2.50°C‰_o^{-1}$ in the ECHAM5-wiso simulation used in Stenni et al. (2017), to be compared to $0.57°C‰_o^{-1}$ and $0.41°C‰_o^{-1}$ for ECHAM5/MPI-OM and $0.78°C‰_o^{-1}$ and $1.51°C‰_o^{-1}$ for the ECHAM5-wiso of Steiger et al. (2017). This clearly shows the sensitivity of the $\delta^{18}$O-temperature relationship.

It is not possible to provide the precise reason explaining the differences observed between models. One cause that may contribute is the time resolution over which the slopes and the correlation coefficients are computed. Indeed, the relationship between $\delta^{18}O$ and surface temperature is dependent -although weakly- of the smoothing applied to the time series. Considering 1- or 10-year averages instead of the 5-year averages used here provide slightly different results (Fig. A2). Those differences are minimal at the continental scale, but this masks variations between regions. Some regions, such the Victoria Land-Ross

Sea, show a decrease of the correlation coefficient and the slope if the size of the bin over which the averages are performed increases, while others, such as the DML coast, show the opposite. But in general, the differences in the $\delta^{18}$O-temperature between 1-, 5- and 10-year averages are small. Besides the fact that the diagnostics could not be computed exactly the same way due to model simulation coverage, the differences in model resolution, as well as the influence of natural variability, may play a role in the differences observed between the simulated $\delta^{18}$O-temperature link

Those results show that the well-known covariance between $\delta^{18}$O and surface temperature is relatively weak, and that it changes over time in ECHAM5/MPI-OM with no apparent link with forcings. Furthermore, it is model simulation- and to some extent smoothing-dependent, but not in a systematic way. If this is a real characteristic applicable to Antarctic isotopes, then this can limit the skill of temperature reconstructions based on statistical methods that rely on a calibration period that may be too short to be representative. It is thus instructive to test a data assimilation method which has the potential advantage

to take into account the temporal changes in the link between temperature and stable isotopes and thus reduce the uncertainties.

## 4.2 Pseudoproxy experiments

This section deals with the potential of reconstructing surface temperature from water stable isotopes, based on pseudoproxy experiments. Using such a controlled framework allows us to assess precisely the performance of the different reconstruction methods through a series of diagnostics including the RMSE and the correlation coefficients between the reconstructions and

the model target (model simulations from which the pseudoproxies are derived), the coefficient of efficiency of the reconstructions and the standard deviation of the model truth and the reconstructions. The coefficient of efficiency (Lorenz, 1956), classically used to measure the skill of reconstructions (e.g. Steiger et al., 2014; Klein and Goosse, 2018), is defined for a time series including $n$ samples as:

$$CE = 1 - \frac{\sum_{i=1}^{n}(x_i - \hat{x}_i)^2}{\sum_{i=1}^{n}(x_i - \bar{x})^2} \qquad (1)$$

where $x$ is the "true" time series, $\bar{x}$ is the "true" time series mean, and $\hat{x}$ is the reconstructed time series, ie. the output after data assimilation or after the statistical reconstruction. CE ranges from one, corresponding to a perfect fit between the "true"





and the reconstructed time series, to $-\infty$. It is negative when the mean of the "true" time series is a better estimate than the reconstructed time series, meaning that the latter has no skill.

The input data for the reconstructions are $\delta^{18}O$ pseudoproxies derived from the ECHAM5/MPI-OM simulation as explained in Section 2.4. Statistical reconstruction methods require to use as input the individual $\delta^{18}O$ pseudoproxies (averaged over a 2°×10° grid) while the related seven Antarctic subregions composites are used as input in data assimilation experiments (see Section 2.4 for more information). It is also possible to assimilate directly the individual $\delta^{18}O$ pseudoproxies. This gives relatively similar reconstructions, although slightly less skilful in our tests. As a consequence, only the results with the assimilation of the seven composites are shown here.

### 4.2.1 Data assimilation of $\delta^{18}O$

The assimilation of $\delta^{18}O$ pseudoproxy derived from ECHAM5/MPI-OM allows as expected to provide reconstructions that are very close to the targets, ie. the ECHAM5/MPI-OM series without any noise added, both when using ECHAM5/MPI-OM and ECHAM5-wiso ensembles (Fig. 5). The RMSE with model truth reach values slightly lower than the errors of the pseudoproxies, which is the best that can be achieved assuming no spatial propagation. The correlation coefficients and the coefficients of efficiency of the reconstructions exceed 0.90 and 0.70 in all regions, except in the Wilkes Land Coast and the Antarctic Peninsula, where a slight decrease in the reconstruction skill is observed in the experiment using the model ensemble derived from ECHAM5-wiso. Overall, ECHAM5-wiso can thus reproduce the temporal and spatial pattern of $\delta^{18}O$ in Antarctica of the $\delta^{18}O$ pseudoproxies derived from ECHAM5/MPI-OM. Despite the strongly different behaviours regarding temperature trends over the last century shown by the two models (Section 3), there is thus no fundamental inconsistencies between them. However, reconstructing $\delta^{18}O$ based on the assimilation of $\delta^{18}O$ is only the minimum requirement to expect skilful reconstructions of temperature.

### 4.2.2 Reconstruction of temperature

Considering Antarctica as a whole, the different methods for reconstructing temperature based on $\delta^{18}O$ pseudoproxies perform similarly well, with correlations with model truth ranging from 0.52 for the data assimilation-based reconstruction using the ECHAM5-wiso ensemble to 0.64 for the data assimilation-based reconstruction using the ECHAM5/MPI-OM ensemble (Fig. 6, all coefficients significant at the 5% level). There is thus some potential of reconstructing the Antarctic temperature based on $\delta^{18}O$ data. However, the temperature reconstruction skill is limited, and is much lower than for the reconstruction of $\delta^{18}O$ (Fig. 5), which once again demonstrates the weak relationship between $\delta^{18}O$ and temperature (Section 4.1).

At the subregional scale, the reconstruction skill generally decreases, and large differences between the reconstruction methods arise. In this case, there is no discernible best reconstruction method, apart from data assimilation using ECHAM5/MPI-OM ensemble that provides the most skilful reconstructions in all regions. This was expected since the $\delta^{18}O$ pseudoproxies used as input for the temperature reconstructions are derived from ECHAM5/MPI-OM, meaning that in this case, the model physics is supposed to be perfect. Those reconstructions using ECHAM5/MPI-OM ensemble are skilful in most individual regions with correlations with temperature model truth ranging from 0.37 in the Antarctic Peninsula to 0.65 in the WAIS, and



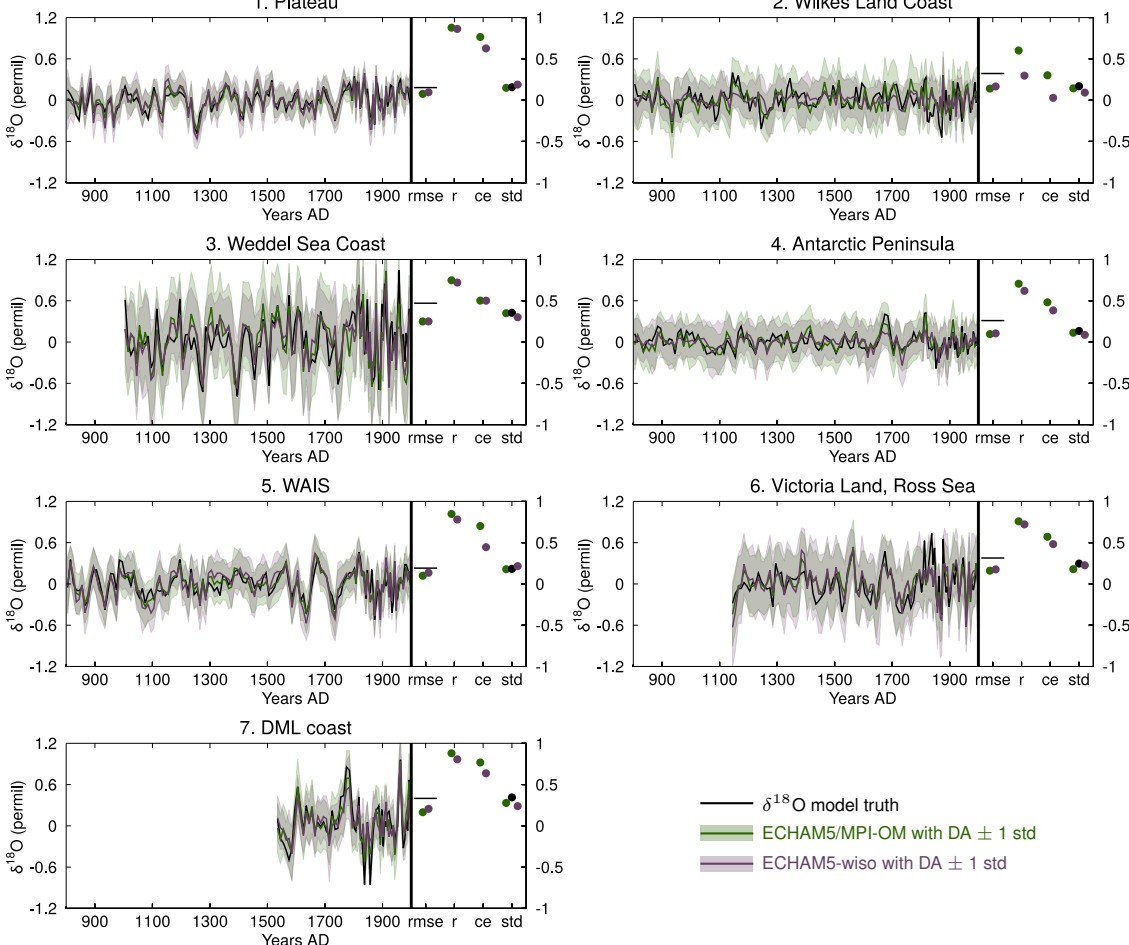

**Figure 5.** Changes in $\delta^{18}O$ in precipitation over the period 800-2000 CE in the model truth (ECHAM5/MPI-OM, from which the pseudoproxies assimilated are derived, in black), and in the data assimilation-based reconstructions using ECHAM5/MPI-OM (in green) and ECHAM5-wiso (in violet) model ensembles. The results are 10-year averaged over the period 800-1800 CE and 5-year averaged over the period 1800-2000 CE. All series are anomalies using the whole periods as a reference. The uncertainty of the reconstructions is shown in shaded area with the corresponding colors (±1 standard deviation of the model particles scaled by their weight around the mean). Diagnostics related to the reconstructions skill are displayed on the right of each panel: the RMSE between the reconstructions and the model target (rmse, in permil) together with the pseudoproxy error estimates (the horizontal black lines), the correlation coefficients between the reconstructions and the model target (r), the coefficient of efficiency of the reconstructions (ce) and the standard deviation of the reconstructions and the model target (std).

positive coefficients of efficiency (Fig. 6). Using the same model for both producing the pseudoproxies and the ensemble of climate states used in the data assimilation process is a strong simplification of reality, that likely artificially inflates the skill of the data assimilation approach (e.g. Smerdon et al., 2016; Klein and Goosse, 2018). Introducing biases in the model physics by





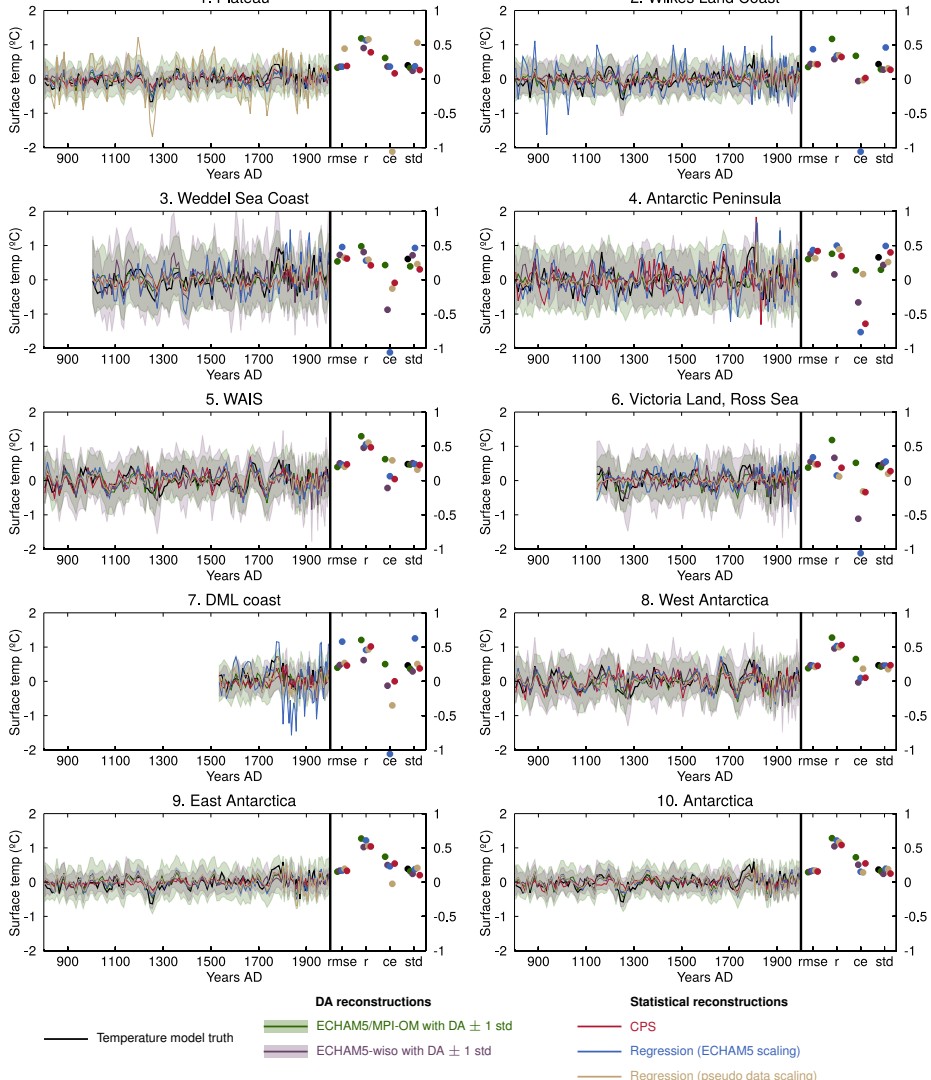

**Figure 6.** Changes in surface temperature over the period 800-2000 CE in the model truth (ECHAM5/MPI-OM, from which the $\delta^{18}$O pseudoproxies assimilated are derived, in black), and in the different data assimilation (in green and violet, see Section 2.3) and statistical (in red, blue and beige, see Section 2.5) reconstructions based on the $\delta^{18}$O pseudoproxies (Fig. 5). The uncertainty of the data assimilation-based reconstructions is shown in shaded area with the corresponding colors ($\pm 1$ standard deviation of the model particles scaled by their weight around the mean). The results are 10-year averaged over the period 800-1800 CE and 5-year averaged over the period 1800-2000 CE. All series are anomalies using the whole periods as a reference. Diagnostics related to the reconstructions skill are displayed on the right of each panel: the RMSE between the reconstructions and the model target (rmse, °C), the correlation coefficients between the reconstructions and the model target (r), the coefficient of efficiency of the reconstructions (ce) and the standard deviation of the reconstructions and the model target (std). The coefficient of efficiency that are lower than $-1$ are displayed just above the label 'ce'.





selecting model states from ECHAM5-wiso to reconstruct temperature shows indeed decreased skills, with correlation coefficients with model truth ranging from 0.08 (not significant at the 5% level) in the Peninsula to 0.48 in the WAIS. The decrease in skill in this case is particularly visible using coefficients of efficiency, with five regions out of seven characterized by negative coefficients. This is not related to a problem in the variance but rather to issues in representing the relative changes. Indeed,

the standard deviation of the series is relatively similar in both data assimilation-based reconstructions for each region, being slightly underestimated compared to the model truth.

The data assimilation reconstructions using the ECHAM5-wiso ensemble are not systematically closer to the model truth than the statistical reconstructions. They are even outperformed in terms of correlation in the Antarctic Peninsula, and, to a lesser extent, in the DML coast region, which may be related to inconsistencies in the model physics and in the spatial structures

compared to the pseudoproxy. More generally, no reconstruction method tends to systematically outperform the others except data assimilation using the ECHAM5/MPI-OM model ensemble. Large discrepancies between the methods skills are observed in all subregions. They are mainly due in statistical reconstructions to differences in the magnitude of the temperature changes, rather than in the relative variability as is the case between both data assimilation-based reconstructions. For instance, on the Plateau, while the statistical reconstruction that is scaled based on the variance of the pseudoproxy over the period 1950-2000

CE has the second highest correlation with the model truth over the last millennium ($r = 0.57$), it has the lowest coefficient of efficiency ($CE = -3.97$) due to a much overestimated variance (standard deviation of the series of $0.53°$ compared to $0.20°$ for the target). This highlights the limits of the hypothesis of the representativeness of short calibration period over longer period. Similar mismatches in variance are observed with the other statistical-based reconstructions. The statistical reconstruction based on the ECHAM5-wiso scaling over the recent past overestimates by a factor two the standard deviation

of the temperature series in the Wilkes Land Coast, by a factor three in the DML coast, and by a factor 1.5 in the Weddell Sea Coast and in the Victoria Land-Ross sea sector. As for the previous reconstruction, this can be related to a calibration period not representative of the past, but also to differences in the $\delta^{18}$O-surface temperature link in the simulation used to scale the reconstructions and in the simulation from which the pseudoproxies are derived. The CPS-based reconstructions are in every subregions in the range of the other reconstruction methods regarding the different diagnostics. As the data assimilation-based

reconstructions, this method can be considered as relatively robust in this case since it does not provide any reconstruction with strongly unrealistic variance, unlike the scaling methods.

## 5   Reconstructions based on real data $\delta^{18}$O

### 5.1   Data assimilation of $\delta^{18}$O

Generally, the data assimilation-based $\delta^{18}$O reconstructions using both models ECHAM5/MPI-OM and ECHAM5-wiso follow

well the trends and the variability shown in the seven regional time series of $\delta^{18}$O presented in Stenni et al. (2017), as indicated by high correlation coefficients, low RMSE close to the data uncertainty, and clearly positive CE (Fig. 7). Since the data assimilation method is built in such a manner that the model physics is respected, the good match means that there are no inconsistencies between measured and simulated spatial patterns and trends in $\delta^{18}$O in precipitation. Only the reconstructions

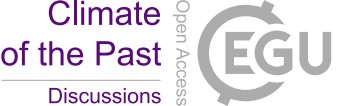



over the regions DML coast and the Weddell Sea Coast show a lower skill, particularly when the model ensemble is derived from ECHAM5-wiso. This is mainly due to an underestimated variance at the decadal scale, related to a limited ensemble size (only 141 model years available in this simulation). Nevertheless, even in this case, the reconstructions still match the assimilated data reasonably well with significant positive correlations and positive CE.

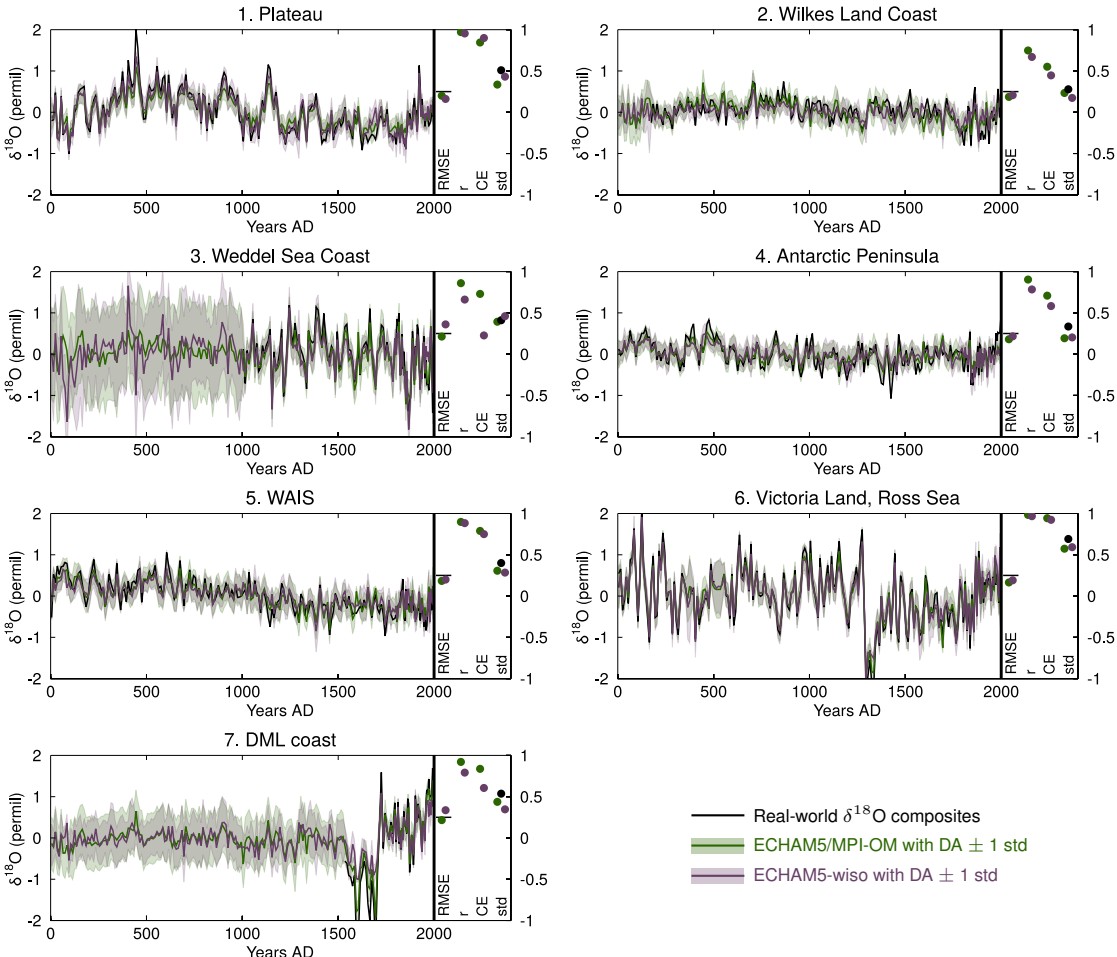

**Figure 7.** Changes in $\delta^{18}$O in precipitation over the period 0-2000 CE in the data assimilated (Stenni et al. (2017), in black), and in the data assimilation-based reconstructions using the model ECHAM5/MPI-OM (in green) and ECHAM5-wiso (in violet). The results are 10-year averaged over the period 800-1800 and 5-year averaged over the period 1800-2000. All series are anomalies using the whole period as a reference. The uncertainty of the reconstructions is shown in shaded area with the corresponding colors ($\pm 1$ standard deviation of the model particles scaled by their weight around the mean). Diagnostics related to the reconstructions skill are displayed on the right of each panel: the RMSE between the reconstructions and the data assimilated (rmse, in permil) together with the data error (the horizontal black lines), the correlation coefficients between the reconstructions and the data assimilated (r), the coefficient of efficiency of the reconstructions (ce) and the standard deviation of the data and the reconstructions over the period covered by the data (std).





From a technical point of view, the data assimilation process works well (see supplementary Section B for technical information). When data are available for assimilation, the uncertainty is reduced meaning that the constraint is strong. When no $\delta^{18}$O data is available, as is the case during the first millennium in the Weddell Sea Coast and the first 1500 years in the DML coast, the uncertainty of the data assimilation-based reconstructions, shown as ±1 standard deviation of the model particles

with non-zero weight around the mean in Fig. 7, is almost as large as the uncertainty of the original model ensembles (not shown). This indicates a very modest influence of the neighbouring regions that have available data to the region lacking data. This nearly total absence of spatial propagation of the information contained in the data assimilated may be related to weak covariances between some of the regions. It seems to be the case at least for the Weddell Sea Coast that is one of the most isolated Antarctic region in terms of interannual variability (Fig. 3). This is consistent with the motivation of Stenni et al.

(2017) to produce regional scale reconstructions.

## 5.2   Reconstruction of temperature

At the continental scale, both the statistical and data assimilation-based reconstructions show a weak warming during the period 0-500 CE, followed by a long-term cooling trend that ends at about the middle of the nineteenth century (Fig. 8). The cooling over the period 850-1850 CE is of the same order of magnitude in the different reconstructions, with values slightly higher

-but still in the range- of the models. Only the reconstruction based on data assimilation using the ECHAM5-wiso ensemble differs from the others, with a weaker cooling over the last millennium, which may be related to the relatively small size of the ensemble and thus of the limited range of possible atmospheric states. At this scale, the variance of the different statistical and data assimilation reconstructions is relatively similar over the period 850-1850 CE.

The main discrepancy between reconstructed temperatures and model results without data assimilation is the too early onset

of the anthropogenic warming, especially visible in West Antarctica. Data assimilation allows reconstructing a later warming that is consistent with statistical reconstructions. After the mid-nineteenth centuries, the reconstructed temperature series are characterized by decadal scale fluctuations with no clear trends, until the mid-twentieth century where the rise in temperature reaches a similar value than the one in reconstruction based on instrumental observations. Over the last 50 years, differences in variance at the continental scale are observed between the time series, but that are not systematically related to the type of

the reconstruction method.

Instrumental observations and models without data assimilation show differences regarding their spatial pattern of the recent trend. The observations clearly display a stronger recent warming in the West than in the East over the past 50 years, while the model mean display a homogeneous warming over Antarctic (Fig.8). All reconstructions, including the data assimilation-based ones, match well the observed contrast between regions. The difference in trend between both regions is however slightly

underestimated in the statistical reconstruction based on the ECHAM5 temperature scaling, and in the data assimilation reconstruction using the model ensemble derived from ECHAM5/MPI-OM. The latter case can be explained by the low range of temperature changes covered by the simulated model states for this reconstruction, despite a high number of particles available (Fig. 1). Nevertheless, data assimilation allows reconciling the dissymmetry of the recent trends between the models ECHAM5/MPI-OM and ECHAM5-wiso and observations, showing that there is no fundamental model-data inconsistency for





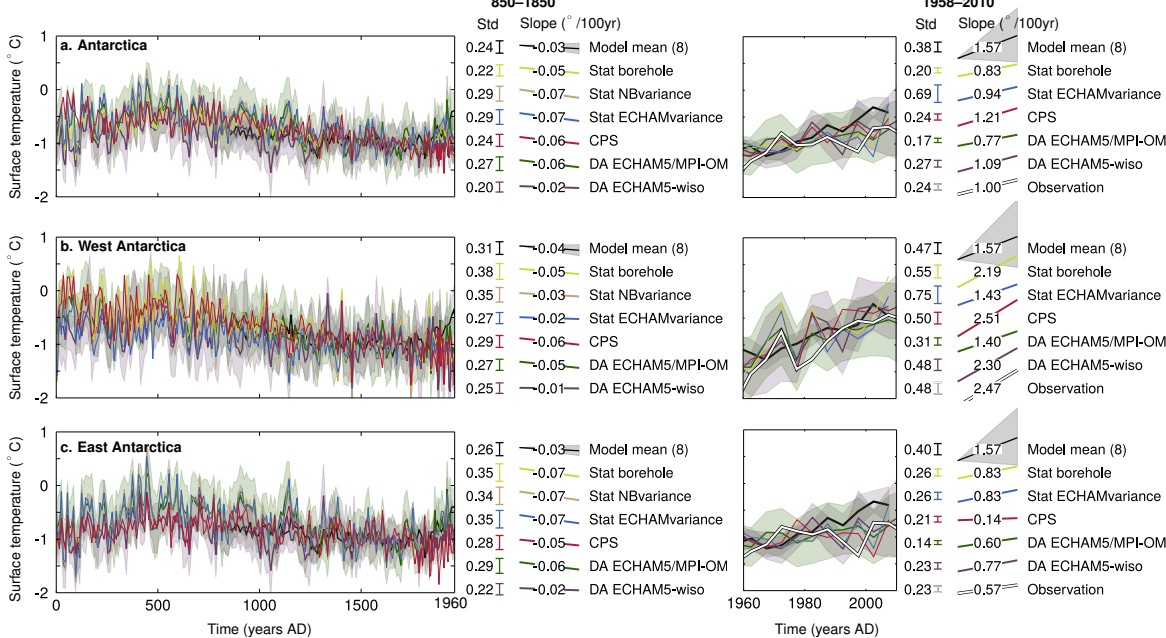

**Figure 8.** Changes in 10- (left panels) and 5-year averaged (right panels) simulated and reconstructed surface temperature over the period 0-2000 CE over a. Antarctica, b. West Antarctica and c. East Antarctica. The model mean ± one standard deviation of the individual simulations is shown in black and in shaded gray, the statistical reconstructions of Stenni et al. (2017) are shown in yellow, beige and blue. Stat NBvariance uses a temperature scaling based on the temperature observations of Nicolas and Bromwich (2014) over the period 1960-2010 CE (see Section 2.5 for more information), Stat ECHAMvariance uses a temperature scaling based on the simulated $\delta^{18}$O-temperature link in a ECHAM5-wiso simulation over the period 1979-2013 CE (Goursaud et al., 2018), and Stat borehole also uses a scaling based on the simulated $\delta^{18}$O-temperature link in ECHAM5-wiso, but the WAIS region is adjusted to match the temperature trend between 1000 and 1600 CE based on borehole temperature measurements Orsi et al. (2012). The data assimilation reconstructions based on the model ensembles ECHAM5/MPI-OM and ECHAM5-wiso are shown in green and violet. The uncertainty of the data assimilation-based reconstructions is shown in shaded area with the corresponding colors (±1 standard deviation of the model particles scaled by their weight around the mean). The reconstructions based on instrumental records observationsNicolas and Bromwich (2014) are shown in white (only in the right panels). The reference period is 1500-1800 CE for the left panels, and 1960-1990 CE for the right panels. The standard deviation and the slopes of the series (in °/100 years) are shown for the periods 850-1850 (or overlap period) and 1958-2010, with drawn bars and slopes proportional to the numbers.

the recent warming pattern in Antarctica, as already suspected by the analysis of all the individual model realizations of the recent trends (Fig. 2-a) and of the recent link between each Antarctica subregions (Fig. 4).

At the large scale (when considering West Antarctica, East Antarctica and Antarctica as a whole), there is thus no strong nor systematic difference in trends, both over the last millennium and over the recent past, when using a data assimilation technique

5  compared to statistical methods applied in Stenni et al. (2017) to reconstruct surface temperature based on water stable isotopes. Furthermore, the variance of the time series produced by the various reconstruction methods are quite similar, although the



data assimilation-based reconstructions often show slightly lower values. The picture becomes different when considering a lower spatial scale with the seven subregions (Fig. 9). In this case, the last millennium trends of the different reconstructions stay relatively similar, although some differences exist, as for instance at the Weddell Sea Coast where the data assimilation reconstructions disagree on the direction of the slope. Note that at this spatial scale, the differences in the model resolution

(3.75° latitude * 3.75° longitude for ECHAM5/MPI-OM and about 1° latitude * 1° longitude for ECHAM5-wiso) may play a role in this disagreement. Unlike the trends, there are large differences in the variance of the last millennium reconstructions. Those differences are found between the data assimilation and the statistical methods, but also among the statistical methods only, while the two data assimilation-based reconstructions show similar variances in every subregions.

Regarding the past 50 years covered by instrumental records, the agreement between the different methods is strongly

region-dependent. For instance, the Plateau is characterized by reconstructions with consistent trends and variances, that show a relatively modest warming and a weak variability. In contrast, in the Victoria Land - Ross Sea sector, the reconstructions based on instrumental observations (Nicolas and Bromwich, 2014) and on data assimilation display a recent warming while the statistical reconstructions show the opposite. There are also differences among similar type of reconstruction methods. For the Weddell Sea Coast, for instance, the statistical reconstructions do not agree on the sign of the recent trend. This also

happens with both data assimilation-based reconstructions in the Wilkes Land Coast and the Antarctic Peninsula. Again, the difference in the model resolution may play a role in this respect. Depending on the region, very large differences in variances can also be found in the produced time series, but, neither one of the statistical-based reconstruction method nor the data assimilation method provide reconstruction whose variance is systematically closer to the one of the instrumental records-based reconstruction. However, one should be careful in drawing conclusions from the analysis of the last 50 years of the

reconstructions, corresponding to the period covered by the instrumental records. This period is indeed very short, especially since only the 5-year mean results are considered, while the seven Antarctic subregions are characterized by a strong variability.

Unlike with the large scale, there are large differences between the reconstruction methods at the subregional scale, mainly in terms of variance, highlighting the uncertainties related to the reconstruction method at this scale. Since the data assimilation technique used can take into account a change in time of the $\delta^{18}$O-surface temperature slope and since this slope does seem

to change strongly over time (Fig. 4), there are, in theory, advantages to use data assimilation over statistical reconstructions. However, whether a variance is better represented in one or another reconstruction cannot be verified given the limited length of the instrumental record. Finally, the main advantage of using data assimilation-based method is that beyond the target variable at the locations where data are available, all variables of the system are reconstructed at all the locations available in the models used. If the reconstruction is consistent with the data assimilated, as is the case here, this allows studying the causes of the

reconstructed changes, although this falls out the scope of the present study.

## 6 Conclusions

The goal of this study is to assess the robustness of Antarctic temperature reconstructions published by Stenni et al. (2017) covering the last two millennia using climate model results and data assimilation experiments. The potential of reconstructing

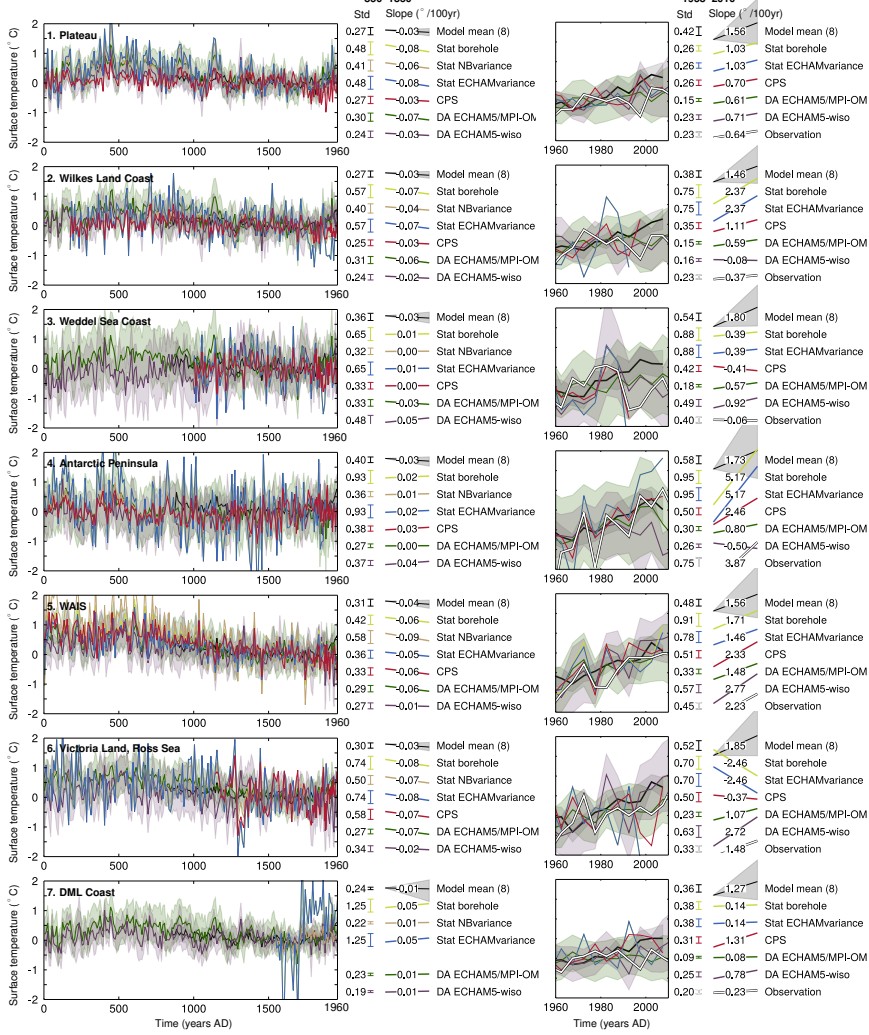

**Figure 9.** Changes in 10- (left panels) and 5-year averaged (right panels) simulated and reconstructed surface temperature over the period 0-2000 CE over the seven Antarctic subregions. The model mean ± one standard deviation of the individual simulations is shown in black and in shaded gray, the statistical reconstructions of Stenni et al. (2017) are shown in yellow, beige and blue. Stat NBvariance uses a temperature scaling based on the temperature observations of Nicolas and Bromwich (2014) over the period 1960-2010 CE (see Section 2.5), Stat ECHAMvariance uses a temperature scaling based on the simulated $\delta^{18}$O-temperature link in a ECHAM5-wiso simulation over the period 1979-2013 CE (Goursaud et al., 2018), and Stat borehole also uses a scaling based on the simulated $\delta^{18}$O-temperature link in ECHAM5-wiso, but the WAIS region is adjusted to match the temperature trend between 1000 and 1600 CE based on borehole temperature measurements Orsi et al. (2012). The data assimilation reconstructions based on the model ensembles ECHAM5/MPI-OM and ECHAM5-wiso are shown in green and violet. The uncertainty of the data assimilation-based reconstructions is shown in shaded area with the corresponding colors (±1 standard deviation of the model particles scaled by their weight around the mean). The reconstructions based on instrumental records observationsNicolas and Bromwich (2014) are shown in white (only in the right panels). The reference period is 1500-1800 CE for the left panels, and 1960-1990 CE for the right panels. The standard deviation and the slopes of the series (in °/100 years) are shown for the periods 850-1850 (or overlap period) and 1958-2010 , with drawn bars and slopes proportional to the numbers.



surface temperature based on water stable isotopes is first examined by characterizing the simulated relationship between both variables through the last millennium in ECHAM5/MPI-OM. The results show that the well-known covariance between $\delta^{18}$O and surface temperature is on average relatively weak. It is characterized by a strong spatial heterogeneity, and changes over time with no apparent link with forcings. Furthermore, the study of the relationship in other isotope-enabled model simulations

from ECHAM5-wiso covering the recent past show that the link differ from one model simulation to another, which may indicate an influence of natural variability in the $\delta^{18}$O-surface temperature link, but also of the model resolution. If those simulated characteristics are real and applicable to Antarctic isotopes, this limits the skill of temperature reconstructions based on statistical methods that rely on the hypothesis that the last decades (the observation period) provide a good estimate for longer temperature reconstructions. Using a data assimilation method to reconstruct temperature based on $\delta^{18}$O has potentially

has thus potentially advantages over statistical methods since it does not rely on a constant $\delta^{18}$O-temperature link through time and space.

Pseudoproxy experiments confirm the benefits of using a data assimilation method, but also the relatively weak link between both variables leading to an only limited potential for reconstructing temperature based on $\delta^{18}$O. No reconstruction method stands out compared to the others in terms of relative variability, but the statistical methods provide reconstructions with

unrealistic variances in some subregions, when the calibration period is too short to provide an adequate estimate of the long term changes of the $\delta^{18}$O-temperature link. In contrast, data assimilation always provide reconstructions with variance in agreement with the model truth. In general, the skill in reconstructing surface temperature based on $\delta^{18}$O data is limited, even in the optimistic framework where the model physics is supposed to be perfect (when assimilating the pseudoproxy derived from ECHAM5/MPI-OM into a model ensemble constructed from ECHAM5/MPI-OM). It is however higher and more uniform

among reconstruction methods when the reconstruction targets are the bigger regions West Antarctica, East Antarctica, and Antarctica as a whole rather than the individual seven subregions.

Applying the data assimilation method to the real $\delta^{18}$O regional composites of Stenni et al. (2017) demonstrates that there is no fundamental model-data inconsistency in terms of temporal and spatial $\delta^{18}$O changes since the output of the assimilation processes using ECHAM5/MPI-OM and ECHAM5-wiso model ensembles match the data assimilated over the seven Antarc-

tic subregions. Consistently with statistical reconstructions, the resulting temperature reconstructions confirm the long term cooling over Antarctica during the last millennium, and the later onset of anthropogenic warming compared to the simulations without data assimilation, especially visible in West Antarctica. Furthermore, data assimilation allows reconciling models and instrumental observations by reconstructing the observed East-West contrast of the recent temperature trends. In instrumental observations, much of the recent warming over Antarctica is indeed due to the strong increase in temperature in West Antarc-

tica, while the East has only weakly warmed over the last 50 years. In contrast, PMIP/CMIP model mean and the mean of the ensemble members of individual PMIP/CMIP models show a uniform warming over both regions following the anthropogenic forcing. Both reconstructions with data assimilation show the observed contrast, indicating that internal variability likely plays a major role in driving this heterogeneous recent warming. This is further supported by the large spread of individual model realizations without data assimilation regarding the spatial pattern of the recent warming. The internal variability is found

to be especially large in West Antarctica, and particularly in the Peninsula. The results of data assimilation experiments and





the analysis of individual model simulations show that the apparent model-data differences for the recent warming pattern in Antarctica do not imply a fundamental inconsistency between models and data. Still, most model simulations show a too homogeneous recent trend compared to data over the continent, but this is not related to a too strong link between regions, the models being able to simulate correlation coefficients between regional temperature changes of the same order as the observed

ones.

Consistently with pseudoproxy experiments results, the temperature reconstructions using the different methods are relatively similar over the three large regions (West Antarctica, East Antarctica, and Antarctica as a whole). At this large scale, there is no large and systematic difference in past and recent trends, nor in magnitude of the variability. This gives credibility to those large scale temperature reconstructions, but it is important to keep in mind that only the uncertainty related to the

reconstruction method is assessed here, and not the potential problems related to the spatial distribution of ice core records, the accuracy of their age scales, and the noise associated with post deposition processes. The picture is different for the seven subregions, where the variance of the last millennium reconstructions produced by the different methods are different. Although there are in theory advantages to using data assimilation over statistical reconstructions that have been confirmed with the pseudoproxy experiments, the instrumental series are too short to confirm it in a realistic setup.

As a perspective, we would like to stress the importance of moving towards the use of a range of climate sensitive proxy records instead of only considering $\delta^{18}$O, given the limited temperature signal present in oxygen isotopes. Assimilating second-order isotopic parameters such as deuterium excess, together with accumulation rates and $\delta^{18}$O would for instance certainly give more insights not just on temperature changes but also on moisture transport characteristics, to help reconstruct hydroclimate variations. However, it seems extremely challenging today since it would require a proper estimation of the observational

error for all proxy records, that is hard to provide. Furthermore, isotope-enabled model simulation covering the last millennium are still rare, despite growing interest in the modelling community (e.g. Werner et al., 2011; Roche, 2013; Werner et al., 2016), and the skill of isotope-enabled models is difficult to assess over the last millennium given the limited availability of instrumental records. Future studies dealing with $\delta^{18}$O data assimilation experiments would take advantage of ensembles of simulations instead of individual runs, to provide a larger range in the simulated states and improve the data assimilation-based

reconstructions skill. More generally, an ensemble of simulations would be useful to any future study involving model-data comparisons of oxygen isotopes over the last millennium, to help distinguish the forced response to natural variability.

*Code and data availability.* The code of the data assimilation method and the resulting Antarctic temperature reconstructions will be made available when the manuscript will be accepted. PMIP3/CMIP5 model results can be downloaded online through the Program for Climate Model Diagnosis and Inter-comparison (PCMDI; http://pcmdi9.llnl.gov). Products from the ECHAM5-wiso model simulation from 1871-

2011 CE (Steiger et al., 2017) can be downloaded from https://doi.org/10.5281/zenodo.1249604.




## Appendix: Supplementary material

## Appendix A: Supplementary figures

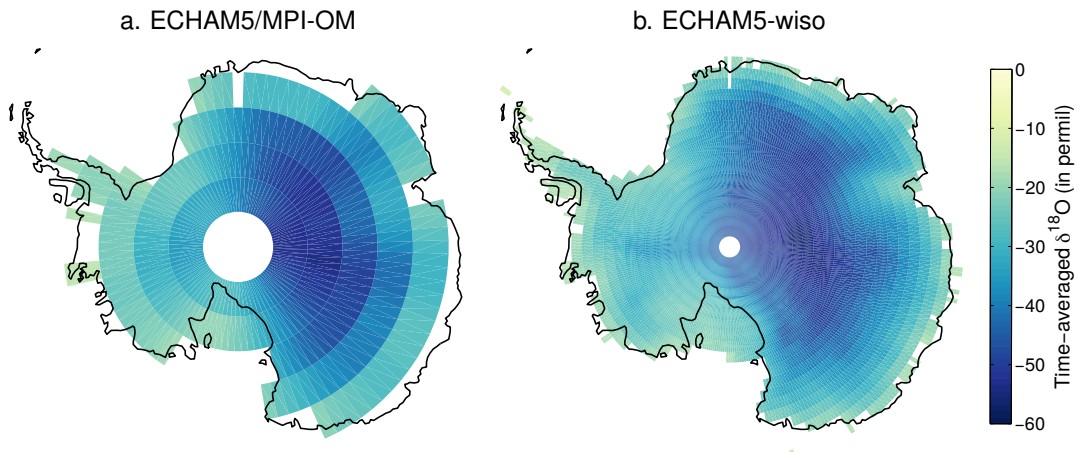

**Figure A1.** Time-averaged $\delta^{18}$O in precipitation in a. ECHAM5/MPI-OM over the period 800-1999 CE and in b. ECHAM5-wiso over the period 1871-2011 CE.

## Appendix B: Information about the data assimilation process

The data assimilation process works well from a technical point of view. The data assimilated constrain strongly the model
5    ensemble, but the constraint is not strong enough to lead to degeneracy, with on average 14% of the particles kept at each data
assimilation step, which makes more than 20 particles out of 141 when using the model ECHAM5-wiso and more than 172 out
of 1200 when using the model simulations performed by ECHAM5/MPI-OM (Fig. B1). The biggest weight given to a particle
reaches in average 11% in the experiment using ECHAM5/MPI-OM and 34% in the experiment using ECHAM5-wiso. The
weights are thus distributed over several model results, indicating the lack of degeneracy and overfitting (Fig. B2).

10   *Competing interests.* The authors declare no competing interests.

*Acknowledgements.* This work was supported by the Belgian Research Action through Interdisciplinary Networks (BRAIN-be) from Belgian
Science Policy Office in the framework of the project "East Antarctic surface mass balance in the Anthropocene: observations and multiscale
modelling (Mass2Ant)" (Contrat n° BR/165/A2/Mass2Ant). H.G. is Research Director within the F.R.S.-FNRS. N.J.S. is supported by US
National Science Foundation award number OISE-1743738. R.N. is supported by the Swiss NSF (PZ00P2_154802).




**Figure A2.** Pearson correlation coefficients (black) and slope (in $°C‰^{-1}$) between surface temperature and $\delta^{18}O$ in ECHAM5/MPI-OM (points) and in ECHAM5-wiso (crosses) over the full period covered by the simulations (800-1999 CE for ECHAM5/MPI-OM and 1871-2011 CE for ECHAM5-wiso), as a function of the bin length over which the data are averaged.

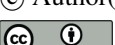



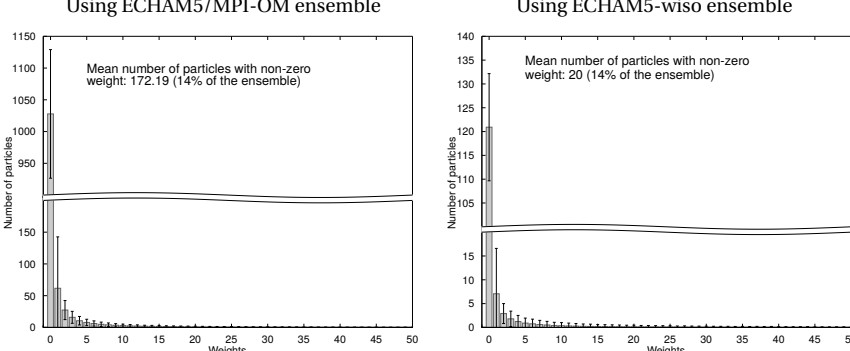

**Figure B1.** Mean distribution of the relative weights (which have to be divided by the total number of particles to obtain the actual weights) of the particles in the reconstructions achieved by the assimilation of the seven composites of $\delta^{18}$O produced in Stenni et al. (2017) using ECHAM5/MPI-OM ensemble (left panel) and using ECHAM5-wiso ensemble (right panel). The distribution is averaged over the period 0-2015 CE, with the standard deviations showed by the error bars.

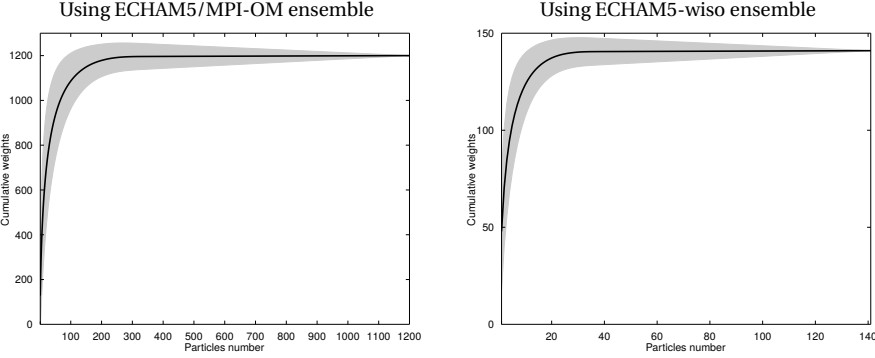

**Figure B2.** Cumulative relative weights (which have to be divided by the total number of particles to obtain the actual weights) of the particles in the reconstructions achieved by the assimilation of the seven composites of $\delta^{18}$O produced in Stenni et al. (2017) using ECHAM5/MPI-OM ensemble (left panel) and using ECHAM5-wiso ensemble (right panel). The particles are sorted in descending order according to their weight. The weights are averaged over the period 0-2015 CE, the shaded area corresponds to plus and minus one standard deviation around the mean.

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
