# Peer review of "Assessing the robustness of Antarctic temperature reconstructions over the past two millennia using pseudoproxy and data assimilation experiments"

_Climate of the Past, 2018_

## Referee Comment (RC1) · Anonymous Referee #1 · 8 Nov 2018

This is an interesting paper that explores important issues (stability of oxygen isotope-temperature relationships) in a novel, thorough and systematic way. Given the challenges of reconstructing temperatures (and climate in general) over the Antarctic, this paper is a valuable step towards better reconstructions, and importantly understanding issues in developing these reconstructions and their uncertainties. The paper is well-written, in particular the conclusions as well as proving a strong summary of what has been found, points well to the implications of the findings, and what is needed to address some of the issues found.

[Figure]

I therefore recommend acceptance subject to minor revisions, which I list below.

1) Abstract, line 3. Consider changing to 'short and spatially sparse' (or something like this) to reflect the issue that the instrumental records are limited in spatial coverage as well as temporal.

2) Page 8 line 33. You discuss that the choice of error has an impact on the results. Some more information on this would be useful (perhaps in supplementary information).

3) Page 9 line 20. You state that the CPS method means that more than half of the records are discarded. Giving the exact number here would be useful.

4) Page 9, line 33. Here and other places where you discuss trends/warming, consider including whether trends are significant.

5) Caption of Figure 1 needs rephrasing. It currently states 'Last millennium 10-(left panels) and 5-year (right panels)', but the right hand panels are not for the last millennium.

6) Caption of Figure 4, line 3, I think should state that the slope values are shown in green.

7) At the start of Section 5, a few sentences reminding the reader of the purpose of this analysis in this section would be helpful. Indeed, doing this at the beginning of each section would help the reader, as the analysis involves quite a few different components/data sources.

8) Small grammatical errors

- Page 2 final line, change to 'also adds to the challenge of the interpretation of ice core signals '

- Page 3 line 26, change to something like 'As our study is based on model results. . . .'
- - page 4 line 1, change to 'consist of using climate model. . .'

- Heading of section 2.2. Change to 'Water stable isotope records'

- page 9, subheading 2.5, change to 'Statistical reconstruction methods' - Page 11 line 16, change 'backyard' to 'backward'

- Page 17 line 19, change to 'there are no fundamental inconsistencies'.

- Caption of figure 8: line 8, change to 'measurements of Orsi et al. (2012). Line 11, rephrase to something like 'The reconstructions based on instrumental records by Nicolas and Bromwich (2014). . .'

- Page 26 line 16, change to 'data assimilation always provides reconstructions'

- Page 27 line 6, change to 'Consistent with the results of the pseudoproxy experiments. . ..'

- Page 27 line 26, change to 'to help distinguish the forced response from natural variability'.

---

## Referee Comment (RC2) · Anonymous Referee #2 · 14 Jan 2019

Review of

'Assessing the robustness of Antarctic temperature reconstructions over the past two millennia using pseudoproxy and data assimilation experiments'

by F. Klein et al.

**Recommendation: minor revisions**

This manuscript presents data assimilation (DA) simulations for Antarctica for the past two millennia using a particle filter with the ECHAM5/MPI-OM and ECHAM5-wiso isotope-enabled GCMs. The simulations comprise pseudo-proxy experiments, which show that the DA can successfully capture the target oxygen isotope pseudoproxies, but that the skill in reproducing temperature variability is limited. It is also shown that this limited skill for temperature reconstructions is due to weak and temporally varying links between regional temperatures and oxygen isotopes, which also means that statistical reconstruction that rely on links fitted during a relatively short period are problematic.

The main objective of the study is to investigate the discrepancies with respect to the beginning of the anthropogenic warming and to the regional temperature trends between statistical temperature reconstructions for Antarctica and forced CMIP simulations. The simulations show an earlier onset and a more spatially homogeneous warming across Antarctica than the empirical temperature reconstructions by Stenni et al. (2017), which shows warming only in some parts including the Antarctic Peninsula and the West Antartic Icesheet. Potential reasons for this mismatch are an overestimation of the forced response in the models, or a dominant role of internal variability. Using assimilation of real-world oxygen isotope records it is shown in the manuscript that the DA simulations are consistent with the empirical temperature reconstructions and that there is therefore no evidence for a fundamental inconsistency between climate simulations and empirical regional Antarctic temperature reconstructions.

The methods applied are state-of-the-art and well explained, and the conclusions are mostly drawn in a sound way. The manuscript is very clearly written and provides an important contribution to palaeoclimate science. There is only one substantial point I would like to be discussed in more detail, which is the distinction of stationary vs transient offline DA methods and the implications on the conclusions. After this and a number of very minor comments have been addressed I fully support the publication of this very interesting and informative paper.

**Specific comments**

1)
There are two types of offline DA methods. In 'transient offline' methods the ensemble used for DA is time-dependent and generated by ensembles of forced simulations, and only the simulated ensemble at or around the time of the assimilation timestep is used for DA. In transient offline DA the ensemble size for DA is limited by the computational constraints on performing transient ensemble simulations. The ensemble size for DA can be substantially increased in 'stationary offline' DA methods by using all simulated timesteps as the ensemble for DA. The transient

offline approach has been used for instance in several studies by Goosse et al., and by Matsikaris et al. (2015); the stationary offline approach has to my knowledge been used the first time by Steiger et al. (2014) and has been applied in several other studies by Steiger et al.

Although it is made clear in the manuscript that a stationary ensemble has been used for DA, the difference between these approaches should be explicitly discussed in section 2.3. Note that the terminology transient/stationary offline is not established yet, but I believe it captures the key difference between the approaches.

Furthermore there should be a discussion on what type of conclusions can be drawn in the two cases if the DA simulations are in agreement with empirical temperature reconstruction. At the moment the conclusion is that there is no fundamental inconsistency between the models and the empirical data. However the question formulated in the introduction was whether the response of the CMIP simulations to the forcing is too strong, or whether internal variability is responsible for the discrepancies between the CMIP simulations and the empirical reconstructions, and the conclusions do not specifically address these two possibilities. In a transient offline approach an agreement between DA simulations and empirical reconstructions would imply that the superposition of forced and internal variability includes the empirically reconstructed states, and thus there is no indication that the forced signal is unrealistic. In contrast when using a stationary offline approach it would be possible to achieve agreement between assimilated states and empirical reconstructions even if the forcing signal was so unrealistic that the superposition of the forced signal and any realistic realisation of internal variability would not include the empirically reconstructed states, because the agreement could be caused by choosing simulated states from times with a different forcing than the actual forcing at a given time.

This shows the limitations of using stationary offline approaches for process studies. The authors' statement 'no fundamental inconsistencies' is fairly vague and a more specific discussion of what is meant by 'fundamental inconsistencies' should be provided.

2)
In section 2.3. it is said that online DA can outperform offline DA when the assimilated data involve a long-term trend. This is just one special case. In general information propagation in time does not have to imply slow changes, as fast changes might still be dynamically related. However, if the system shows slow changes it is clear that information is propagated forward in time. The explanations should be adjusted accordingly.

3)
Page2, line 17, replace 'signal' with 'change'

4)
Page 3, line 26, 'Our study being based … it is important'; wrong English

5)
Page5, lines 14/15, ' ... simulate similar … than another …', not well phrased, either replace 'than' with 'as' or reformulate.

6)
Page5, line 19, replace 'validating' with 'justifying'

7)
Page 8, line 1, replace 'of' with 'for'

8)
Page 8, line 9, replace 'on' with 'to'

9)
Page 8, line 18, replace 'pseudoproxy' with 'pseudoproxies'

10)
Page 11, line 8, replace 'simulation' with 'simulations'

11)
Page 12, line 11, replace 'model mean' with 'model mean correlation' (if I understand correctly)

13)
Page 14, line 1, replace 'of' with 'for'

14)
Page 14, line 25/26 'in the results with a last century …', something is wrong with this sentence

15)
Page 14, line 28, replace 'link between' with 'links of'

16)
Page 20, line 17, replace 'hypothesis' with 'assumption'

17)
Page 26, line 9, delete 'has potentially'

---

## Author Comment (AC1) · 11 Feb 2019

**Answer to referee 1**

The referee's comments are shown in black and our answers in blue :

This is an interesting paper that explores important issues (stability of oxygen isotope-temperature relationships) in a novel, thorough and systematic way. Given the challenges of reconstructing temperatures (and climate in general) over the Antarctic, this paper is a valuable step towards better reconstructions, and importantly understanding issues in developing these reconstructions and their uncertainties. The paper is well-written, in particular the conclusions as well as proving a strong summary of what has been found, points well to the implications of the findings, and what is needed to address some of the issues found.

I therefore recommend acceptance subject to minor revisions, which I list below.

We would like to thank the reviewer for the positive evaluation of our manuscript and for the useful comments that will be addressed in the revised version as specified here :

1) Abstract, line 3. Consider changing to 'short and spatially sparse' (or something like this) to reflect the issue that the instrumental records are limited in spatial coverage as well as temporal.

This will be modified accordingly.

2) Page 8 line 33. You discuss that the choice of error has an impact on the results. Some more information on this would be useful (perhaps in supplementary information).

We agree with the reviewer that this is an important point. We thus plan to change this sentence (p8l32) :

*Those different estimates of the data uncertainty have an impact on the results, but no best choice could be determined based on our experiments.*

by:

*Those different estimates of the data uncertainty have an impact on the results, but, as shown in the supplementary Section B, this impact is limited in our experiments.*

and to add a new supplementary section (including a new figure) discussing the choice of the data error:

**Appendix B: Data uncertainty sensitivity**

*Specifying the error on the data is a key element of the data assimilation process. The smaller it is, the stronger the constraint provided by the data will be. In paleoclimatology, obtaining the right estimate of this error is challenging as there is often no quantitative uncertainty provided with data series derived*

*from observation. This implies that we have to make a choice that will inherently be associated with strong hypotheses or subjective considerations.*

*As mentioned in Section 2.4, several distributions of data error have been considered here. First, the uncertainty has been assumed spatially homogeneous, meaning that each data series has the same error (either 0.15‰, 0.25‰ or 0.50‰). While this strategy is likely unrealistic, it has the advantage to be very simple. The data error has also been considered proportional to the variance of the data series. It implies the same signal to noise ratio in all the series but there is no obvious reason to justify this hypothesis. Finally, the data error has been defined as proportional to the regression error term between observed temperature and $\delta^{18}O$. However, we have shown here that the temperature-$\delta^{18}O$ link is weak and temporally and spatially varying. Furthermore, the time overlap between the temperature observations (Nicolas and Bromwich, 2014), covering the past 50 years, and the $\delta^{18}O$ data (Stenni et al., 2017), often missing the recent past, is very short. Given the 5-year temporal resolution of the $\delta^{18}O$ series (see Section 2.2 for more information), the regression is computed using maximum 10 points, hampering its reliability.*

*To limit as much as possible choices that may be hard to justify, the data assimilation-based reconstructions analyzed in this paper have used a spatially homogeneous uncertainty of 0.25‰. Fortunately, it appears that the different strategies used to estimate the uncertainty of the data give regional temperature reconstructions that are relatively consistent (Fig. B1). Although there are some weak differences in variance, the different reconstructions show similar patterns over the last two millennia. This relatively limited impact on the results of the way the data error is estimated adds robustness to the reconstructions based on data assimilation.*

[Figure]

*Figure B1: Data assimilation-based temperature reconstructions using the model ensemble ECHAM5/MPI-OM over the period 0-2000 CE over the seven Antarctic subregions. The time series differ due to different data error taking into account in the data assimilation process: 0.25‰ spatially homogeneous (in green), 0.50‰ spatially constant (in gray), 0.5 × the standard deviation of the data series (in blue), and 0.5× the residual sum of squares from the linear regression predicting observed temperature from reconstructed $\delta^{18}O$ (in red). The uncertainty of the reconstructions is shown in shaded area with the corresponding colors (±1 standard deviation of the model particles scaled by their weight around the mean). The reference period is 1500-1800 CE.* **Attention, note that the symbols 'permil' has been changed to 'h' in this document. This will of course not be the case in the revised version of the manuscript.**

3) Page 9 line 20. You state that the CPS method means that more than half of the records are discarded. Giving the exact number here would be useful.

This will be specified (62 out of 112).

4) Page 9, line 33. Here and other places where you discuss trends/warming, consider including whether trends are significant.

We will specify the significance of F tests (p-value<0.05) testing for a non-zero slope everywhere on the text where numbers of trends are given, as well as in the Figures 1, 8 and 9 where the slope values will be followed by a star (*) when significant.

5) Caption of Figure 1 needs rephrasing. It currently states 'Last millennium 10- (left panels) and 5-year (right panels)', but the right hand panels are not for the last millennium.

It will be modified :

*Changes in 10- (left panels) and 5-year averaged (right panels) surface temperature over the period 850-2000 CE ...*

6) Caption of Figure 4, line 3, I think should state that the slope values are shown in green.

It will be added in the revised version.

7) At the start of Section 5, a few sentences reminding the reader of the purpose of this analysis in this section would be helpful. Indeed, doing this at the beginning of each section would help the reader, as the analysis involves quite a few different components/data sources.

Thank you for the remark. We will add at the start of Section 5:

*In the same way as for the pseudoproxy experiments, we first assess here whether model results can match in the data assimilation experiments the $\delta^{18}O$ reconstructions of Stenni et al. (2017), which are based on ice core measurements. This is needed to potentially obtain skillful temperature reconstructions. However, given the relatively weak link between $\delta^{18}O$ and temperature evidenced in Sections 4.1 and 4.2, the skill of those temperature reconstructions is expected to be limited even if the data assimilation process technically works well.*

8) Small grammatical errors
We would like to thank the reviewer for noticing all the following typing and spelling errors that will be corrected in the revised version of the manuscript.

- Page 2 final line, change to 'also adds to the challenge of the interpretation of ice core signals '
This will be changed.

- Page 3 line 26, change to something like 'As our study is based on model results . . .'
This will be changed as proposed.

- Page 4 line 1, change to 'consist of using climate model . . .'
This will be changed as modified accordingly.

- Heading of section 2.2. Change to 'Water stable isotope records'
This will be changed.

- page 9, subheading 2.5, change to 'Statistical reconstruction methods'
This will be changed.

- Page 11 line 16, change 'backyard' to 'backward'
This will be changed.

- Page 17 line 19, change to 'there are no fundamental inconsistencies'.
This will be changed.

- Caption of figure 8: line 8, change to 'measurements of Orsi et al. (2012). Line 11, rephrase to something like 'The reconstructions based on instrumental records by Nicolas and Bromwich (2014)… '
Thank you for noticing. This will be changed accordingly.

- Page 26 line 16, change to 'data assimilation always provides reconstructions'
This will be changed.

- Page 27 line 6, change to 'Consistent with the results of the pseudoproxy experiments....'
This will be modified.

- Page 27 line 26, change to 'to help distinguish the forced response from natural variability'.
This will be modified.

---

## Author Comment (AC2) · 11 Feb 2019

**Answer to referee 2**

The referee's comments are shown in black and our answers in blue :

**Review of**

'Assessing the robustness of Antarctic temperature reconstructions over the past two millennia using pseudoproxy and data assimilation experiments'

by F. Klein et al.

**Recommendation: minor revisions**

This manuscript presents data assimilation (DA) simulations for Antarctica for the past two millennia using a particle filter with the ECHAM5/MPI-OM and ECHAM5-wiso isotope-enabled GCMs. The simulations comprise pseudo-proxy experiments, which show that the DA can successfully capture the target oxygen isotope pseudoproxies, but that the skill in reproducing temperature variability is limited. It is also shown that this limited skill for temperature reconstructions is due to weak and temporally varying links between regional temperatures and oxygen isotopes, which also means that statistical reconstruction that rely on links fitted during a relatively short period are problematic.

The main objective of the study is to investigate the discrepancies with respect to the beginning of the anthropogenic warming and to the regional temperature trends between statistical temperature reconstructions for Antarctica and forced CMIP simulations. The simulations show an earlier onset and a more spatially homogeneous warming across Antarctica than the empirical temperature reconstructions by Stenni et al. (2017), which shows warming only in some parts including the Antarctic Peninsula and the West Antactic Icesheet. Potential reasons for this are an overestimation of the forced response in the models, or a dominant role of internal variability. Using assimilation of real world oxygen isotope records it is shown in the manuscript that the DA simulations are consistent with the empirical temperature reconstructions and that there is therefore no evidence for a fundamental inconsistency between climate simulations and empirical regional Antarctic temperature reconstructions.

The methods applied are state-of-the-art and well explained, and the conclusions are mostly drawn in a sound way. The manuscript is very clearly written and provides an important contribution to palaeoclimate science. There is only one substantial point I would like to be discussed in more detail, which is the distinction of stationary vs transient offline DA methods and the implications on the conclusions. After this and a number of very minor comments have been addressed I fully support the publication of this very interesting and informative paper.

We would like to warmly thank the reviewer for his careful evaluation of our manuscript and for the interesting comments. All of them will be taken into account in the revised version.

**Specific comments**

1)

There are two types of offline DA methods. In 'transient offline' methods the ensemble used for DA is time-dependent and generated by ensembles of forced simulations, and only the simulated ensemble at or around the time of the assimilation timestep is used DA. In transient offline DA the ensemble size for DA is limited by the computational constraints on performing transient ensemble simulations. The ensemble size for DA can be substantially increased in 'stationary offline' DA methods by using all simulated timesteps as the ensemble for DA. The transient offline approach has been used for instance in several studies by Goosse et al., and by Matsikaris et al. (2015); the stationary offline approach has to my knowledge been used the first time by Steiger et al. (2014) and has been applied in several other studies by Steiger et al.

Although it is made clear in the manuscript that a stationary ensemble has been used for DA, the difference between these approaches should be explicitly discussed in section 2.3. Note that the terminology transient/stationary offline is not established yet, but I believe it captures the key difference between the approaches.

Thank you for the remark. We agree that it is important and will thus include in Section 2.3 a paragraph about the difference between those two methods:

There are two types of offline data assimilation methods which differ by the way the model ensembles are produced. They can be referred to as transient and stationary offline methods. In transient methods (e.g. Goosse et al., 2006; Bhend et al., 2012; Matsikaris et al., 2015), an ensemble of simulations is first generated by performing several simulations with one model driven by realistic estimates of the forcing. The ensemble of states used for the data assimilation (i.e. the prior) is time-dependent and changes at every assimilation step since the model results and the data must correspond to the same time (generally the same year). As for online methods, transient offline methods have the advantage to provide reconstructions that are consistent with changes in forcings. However, obtaining skillful reconstructions depends on the range of the ensemble that must be wide enough to capture the full complexity included in the data network. This is directly related to the ensemble size, which is strongly limited in transient offline methods by the computational constraints on performing ensemble simulations. In stationary offline methods (e.g. Steiger et al., 2014; Hakim et al., 2016; Steiger et al., 2018), the ensemble of states used for the data assimilation is obtained by selecting not only the time in the simulations corresponding to the data assimilation time step (and thus the observed changes) but also other simulated time steps. This allows increasing the ensemble size by several orders of magnitude and thus potentially the skill of the reconstructions. However, since the prior includes years with many different forcings, the resulting reconstructions may be inconsistent with changes in the forcing history. This is still valid when internal variability dominates over the forced response, as is the case for instance with hydroclimate-related variables at local scale (e.g. Klein and Goosse, 2018). If the fingerprint of the forcing is large, the data assimilation procedure can also select for the reconstruction during a specific year only simulated years with a forcing similar to the one observed during that year. However, it is also possible

**that the forcing contribution is underestimated in the reconstruction due to the selection of the prior inducing some different teleconnections compared to the observed ones and troubles in the interpretation of the reconstructed patterns.**

Furthermore there should be a discussion on what type of conclusions can be drawn in the two cases if the DA simulations are in agreement with empirical temperature reconstruction. At the moment the conclusion is that there is no fundamental inconsistency between the models and the empirical data. However the guestion formulated in the introduction was whether the response of the CMIP simulations to the forcing is too strong, or whether internal variability is responsible for the discrepancies between the CMIP simulations and the empirical reconstructions, and the conclusions do not specifically address these two possibilities. In a transient offline approach an agreement between DA simulations and empirical reconstructions would imply that the superposition of forced and internal variability includes the empirically reconstructed states, and thus there is no indication that the forced signal is unrealistic. In contrast when using a stationary offline approach it would be possible to achieve agreement between assimilated states and empirical reconstructions even if the forcing signal was so unrealistic that the superposition of the forced signal and any realistic realisation of internal variability would not include the empirically reconstructed states, because the agreement could be caused by choosing simulated states from times with a different forcing than the actual forcing at a given time.

This shows the limitations of using stationary offline approaches for process studies. The authors' statement 'no fundamental inconsistencies' is fairly vague and a more specific discussion of what is meant by 'fundamental inconsistencies' should be provided.

We concur with this comment and with the importance of specifying what can actually be done or not with such DA method. Several changes will accordingly be made in the manuscript. In sequence starting by the abstract :

A/ For the reasons you mentioned, this is not because our DA-based reconstruction match the observed recent trend that we can state that it is driven by internal variability. Hence, the following sentence (p2l8-11) :

Data assimilation also allows reconciling models and direct observations by reconstructing the East-West contrast regarding the recent temperature trends, indicating that internal variability likely plays a major role in driving this heterogeneous recent warming. This is further supported by the large spread of individual PMIP/CMIP model realizations regarding the recent warming pattern.

will be replaced by :

Data assimilation also allows reconciling models and direct observations by reproducing the East-West contrast in the recent temperature trends. This recent warming pattern is likely mostly driven by internal variability given the large spread of individual PMIP/CMIP model realizations in simulating it. B/ We will make clear when describing the DA method (Section 2.3) that using a stationary method is not ideal to study the processes responsible for the reconstructed changes (which is out of scope of our study) since they can be the results of a mix between several forced and internal variability-based influences (see the proposed paragraph to be included in Section 2.3 above).

C/ We propose to be more specific when stating in Section 5.2 that there are no fundamental inconsistencies between models and observations about the recent warming pattern (p22I34). From p22I33:

Nevertheless, data assimilation allows reconciling the apparent disagreement on the recent trends between the models ECHAM5/MPI-OM and ECHAM5-wiso and observations. We use a stationary offline data assimilation method. This means that when all simulated years are analyzed, models can simulate a pattern resembling the observed contrasted warming between East and West Antarctica. This implies that such pattern is consistent with model physics and that internal variability has likely a strong role in the this observed pattern, as suggested by the analysis of all the individual model realizations of the recent trends (Fig. 2-a) and of the recent link between each Antarctic subregions (Fig. 4). However, because of our experimental design, there is no guarantee that the contribution of the forcings is well taken into account. For instance, we cannot rule out that although the pattern is compatible with internal variability, it cannot be totally masked in some models by a too strong response to the forcing leading to an incompatibility with observations.

D/ We also propose to slightly change the conclusions. These sentences (p26l32) :

Both reconstructions with data assimilation show the observed contrast, indicating that internal variability likely plays a major role in driving this heterogeneous recent warming. This is further supported by the large spread of individual model realizations without data assimilation regarding the spatial pattern of the recent warming.

**will be replaced by:**

Both reconstructions with data assimilation show the observed contrast, indicating that this pattern can be represented by climate models. Furthermore, the large spread of individual model realizations without data assimilation regarding the spatial pattern of the recent warming suggests that internal variability likely plays a major role in driving this heterogeneous recent warming.

**2)**

In section 2.3. it is said that online DA can outperform offline DA when the assimilated data involve a long-term trend. This is just one special case. In general information propagation in time does not have to imply slow changes, as fast changes might still be dynamically related.

However, if the system shows slow changes it is clear that information is propagated forward in time. The explanations should be adjusted accordingly.

Thank you for the comment. We agree that the explanation was not precise enough and we propose to change (p7I7):

An online method can theoretically outperform an offline one when the data assimilated involves a long-term trend since some components of the climate system can propagate information forward in time from one assimilation step to the next one (Pendergrass et al., 2012; Matsikaris et al., 2015).

by:

An online method can theoretically outperform an offline one if the state of the system at one particular time significantly influences its subsequent evolution, as it allows the propagation of the information forward in time from one assimilation step to the next one (Pendergrass et al., 2012; Matsikaris et al., 2015).

Thank you for noticing all the following typing and spelling errors that will be corrected:

3)

Page2, line 17, replace 'signal' with 'change' This will be modified.

4)

Page 3, line 26, 'Our study being based ... it is important'; wrong English This will be changed to 'As our study is based on model results, ...'

5)

Page5, lines 14/15, ' ... simulate similar ... than another ...', not well phrased, either replace 'than' with 'as' or reformulate. 'than' will be replaced by 'as'.

6) Page5, line 19, replace 'validating' with 'justifying' This will be replaced.

7) Page 8, line 1, replace 'of' with 'for This will be replaced.

8)

Page 8, line 9, replace 'on' with 'to' This will be replaced.

9) Page 8, line 18, replace 'pseudoproxy' with 'pseudoproxies' This will be replaced.

10) Page 11, line 8, replace 'simulation' with 'simulations' This will be replaced.

**11)**

Page 12, line 11, replace 'model mean' with 'model mean correlation' (if I understand correctly) The sentence will be modified for more clarity:

The simulated link between East and West Antarctica is rather consistent for each model and similar to the observed one, as deduced from correlation coefficients computed using the mean of all members for each model.

**13) Page 14, line 1, replace 'of' with 'for'**

This will be modified.

**14)**

Page 14, line 25/26 'in the results with a last century...', something is wrong with this sentence

Thank you for noticing. 'with' will be replaced by 'showing'.

15)

Page 14, line 28, replace 'link between' with 'links of' This will be modified.

16)

Page 20, line 17, replace 'hypothesis' with 'assumption' This will be replaced.

17)

Page 26, line 9, delete 'has potentially' This will be deleted.